



# Source apportionment resolved by time-of-day for improved deconvolution of primary source contributions to air pollution

Sahil Bhandari[1], Zainab Arub[2], Gazala Habib[2], Joshua S. Apte[3,4], and Lea Hildebrandt Ruiz[5]

[1]Department of Mechanical Engineering, University of British Columbia, Vancouver, Canada
[2]Department of Civil Engineering, Indian Institute of Technology Delhi, New Delhi, India
[3]Department of Civil and Environmental Engineering, UC Berkeley, California, USA
[4]School of Public Health, UC Berkeley, California, USA
[5]McKetta Department of Chemical Engineering, The University of Texas at Austin, Texas, USA

*Correspondence to*: Lea Hildebrandt Ruiz (lhr@che.utexas.edu)

**Abstract.** Present methodologies for source apportionment assume fixed source profiles. Since meteorology and human activity patterns change seasonally and diurnally, application of source apportionment techniques to shorter rather than longer time periods generates more representative mass spectra. Here, we present a new method to conduct source apportionment resolved by time of day using the underlying approach of positive matrix factorization (PMF). We call this approach "time-of-day PMF" and statistically demonstrate the improvements in this approach over traditional PMF. We report on source
apportionment conducted on four example time periods in two seasons (winter and monsoon 2017), using organic aerosol measurements from an Aerosol Chemical Speciation Monitor (ACSM). We deploy the EPA PMF tool with the underlying Multilinear Engine (ME-2) as the PMF solver. Compared to the traditional seasonal PMF approach, we extract a larger number of factors as well as PMF factors that represent the expected sources of primary organic aerosol using time-of-day PMF. By capturing diurnal time series patterns of sources at a low computational cost, time-of-day PMF can utilize large datasets
collected using long-term monitoring and improve the characterization of sources of organic aerosol compared to traditional PMF approaches that do not resolve by time of day.

## 1 Introduction

Air pollution is considered the greatest current environmental health threat to humanity, with an estimated mortality burden of 7 million per year (World Health Organization, 2018; Schraufnagel et al., 2019; Health Effects Institute, 2020). Air pollutants
also cause climate forcing and environmental damages to ecosystems and biodiversity (Intergovernmental Panel on Climate Change, 2019; Intergovernmental Panel on Climate Change, 2021). Apart from physiological and environmental effects, air pollution is associated with negative psychological, economic, and social effects (Lu et al., 2020). High racial, ethnic, income, regional, and nationality-based disparities exist in air pollution exposure, making air pollution exposure an important environmental justice issue (Hajat et al., 2015; Goodkind et al., 2019; Tessum et al., 2019; Thind et al., 2019; Health Effects
Institute, 2020; Pandey et al., 2020; Chakraborty et al., 2021). These disparities are associated with a wide variety of sectors,





activities, processes, and pollutants (Thakrar et al., 2020). Policy solutions targeting specific pollutants have led to non-uniform reductions of air pollution contributions of different sectors (Tschofen et al., 2019). Thus, reduction of air pollution is essential to global health and can be expected to generate long-term societal benefits (Tessum et al., 2018; Tschofen et al., 2019; Organization for Economic Co-operation and Development, 2020). However, more than half the world's population is exposed
to increasing air pollution (Shaddick et al., 2020). Most of this population lives in developing nations. Moreover, economic resources are limited, and reduction of air pollution alongside continued economic growth requires investment in abatement measures for older technologies and adoption of cleaner technologies (Lei et al., 2021). Thus, sources of air pollution need to be prioritized to appropriately focus limited resources on the most effective abatement measures. This prioritization should be based on the contributions of different emission sources to air pollution in a region.

Source apportionment is the practice of attributing air pollution to different causes such as sectors (residential, industrial), activities (traffic, biomass burning) and atmospheric processes (oxidation). Several approaches have been developed to conduct source apportionment studies (Belis et al., 2014). Broadly, these approaches can be categorized into emission inventories, receptor-oriented modeling, and source-oriented modeling. These approaches have been accepted by regional, national, and international agencies for use in air quality policy and planning (Belis et al., 2014; Environmental
Protection Agency, 2017; California Air Resources Board, 2018; Wayland, 2018). Source-oriented models and emission inventories together capture the emissions, chemical transformation, transport, and dispersion of pollution. However, they have heavy computational burden, require extensive data collection, and are subject to cumulative uncertainties from model inputs as well as the different computational components (Hopke et al., 2016). Receptor models are mathematical tools with relatively lower computational requirements that use mass balance analysis to output source contributions (time series of source
concentrations) and source profiles (relative strength of different pollutants) for identified sources of air pollution (Belis et al., 2013; Hopke et al., 2016). Positive Matrix Factorization (PMF) has been identified as an appropriate receptor modeling technique that can be deployed for quantifying source contributions for air quality management (Belis et al., 2015).

      Three tools are currently in active use for application of PMF to atmospheric datasets: the IGOR PMF Evaluation Tool (PET) (Ulbrich et al., 2009), the EPA PMF tool (Brown et al., 2012), and Source Finder (SoFi) (Canonaco et al., 2013).
The IGOR PET tool uses the PMF2 program to resolve factors from 2-D matrices (Paatero and Tapper, 1994; Ulbrich et al., 2009). Further details on the statistical basis of this method are available elsewhere (Ulbrich et al., 2009; Zhang et al., 2011, and references therein). Both SoFi and EPA PMF are based on the Multilinear Engine (ME-2), which allows the application of factor profile constraints to extract specific sources (Paatero et al., 1999; Paatero et al., 2002; Canonaco et al., 2013; Crippa et al., 2014; Norris et al., 2014). PMF2 does not allow the application of factor profile constraints, and it often results in greater
uncertainty in solutions, poorer source separation, and fewer identified sources compared to ME-2 (Ramadan et al., 2003, Amato et al., 2009; Amato and Hopke, 2012). A further important advantage of the EPA PMF tool over Igor PET and SoFi are its error estimation techniques, which systematically account for both random error and rotational ambiguity using bootstrapping, displacements, and bootstrap-displacements, as explained in more detail in Sect. 2.5. (Paatero et al., 2014;



Brown et al., 2015). Currently, the IGOR PET and the SoFi tools only use bootstrapping to account for random errors and,
partially, rotational ambiguity (Ulbrich et al., 2009; Canonaco et al., 2020).

      PMF tools have been applied to identify sources using long-term datasets spanning multiple years (Zhang et al., 2019;
Heikkinen et al., 2020), seasonal datasets accounting for seasonal variability (Amil et al., 2016; Bikkina et al., 2019; Bhandari
et al., 2020; Patel et al., 2021a), for studying special events (Reyes-Villegas et al., 2018; Rai et al., 2020, Patel et al., 2021b),
to study spatial variability (Crippa et al., 2014; Robinson et al., 2018), as well as to connect sources to health effects
(Daellenbach et al., 2020). Several studies have analysed the influence of meteorology after conducting source apportionment
on a larger dataset (Venturini et al., 2014; Pauraite et al., 2019; Bhandari et al., 2020). Some studies have quantified the effect
of meteorological variables on the performance of the source apportionment approach for the identification of sources, with or
without stratification. One such study stratified data based on mean temperature and showed that accounting for temperature
variability using gas-particle partitioning before conducting source apportionment improved the stability of the solution (Xie
et al., 2013a, b). Similar data-segmentation schemes have been deployed for wind direction, wind speed, and precipitation, and
these techniques resulted in a larger and more representative PMF factors (Park et al., 2019).

      A major limitation of PMF is the assumption of constant factor profiles throughout the modeling period—while the
contribution of each factor is modelled to change over time, its profile (e.g., mass spectrum, when PMF is applied to mass
spectrometer data) stays constant, which leads to modeling uncertainty (Ulbrich et al., 2009). Previous studies have tested the
limitation of constant mass spectral profiles for seasonal and weekly changes in meteorology and activity patterns (Canonaco
et al., 2015; Reyes-Villegas et al., 2016; Canonaco et al., 2020). These studies found that annual and seasonal datasets from
an Aerosol Chemical Speciation Monitor (ACSM; Aerodyne Research, Billerica, MA) show high variations in mass spectral
contributions which cannot be sufficiently captured when PMF is conducted on the complete dataset. These studies
recommended conducting PMF analysis on shorter time frames (weeks–months) with limited variability of emissions and
meteorology. However, meteorological conditions influence source apportionment on hourly and smaller time scales; for
example, changes in ventilation (Dai et al., 2020) and photochemistry (Lelieveld et al., 1991) affect source apportionment
results. Human activity patterns also vary with time leading to changes in source cocktails—for example, we expect higher
cooking emissions during cooking-influenced periods (Abdullahi et al., 2013; Patel et al., 2021), higher traffic emissions during
rush hours (Zhang et al., 2013), and time-of-day, day-of-week, and month-of-year patterns for other emission sources (Crippa
et al., 2020). These changes in meteorology, photochemistry, and sources lead to diurnal variability in mass spectra. For
example, Canonaco and co-workers (2015) showed that the mass spectra of secondary organic aerosol (SOA) changed with
concentrations of OX (O3+NO2), which shows high diurnal variability due to monotonic association with ambient
temperature. Finally, diurnal variability of time series patterns is frequently used for PMF factor selection and representation
(Zhang et al., 2011). As an example, using data from 11 days of PMF runs, Williams and co-workers presented bihourly diurnal
variability of PMF factor time series contributions (Williams et al., 2010). Recognizing the importance of variability of source
influence at receptor sites, previous research has examined the influence of sampling periods, sampling time resolution, and
time series variability of source emissions on the final PMF result (Tian et al., 2017; Wang et al., 2018). Results from these


studies suggest that, given the assumption of constant factor profiles in PMF, PMF analysis should be conducted on time-resolved datasets. Additionally, to capture source emission and meteorological variability over the day, data from all times of
day should be collected. Thus, an ideal PMF technique would make the most of the high-time resolution of datasets while assuming constant factor profiles for periods with limited variability in emissions and meteorology.

One such approach is conducting different PMF runs for different times of the day across long-term datasets. A key advantage of such sub-setting is that it captures the diurnal variability in source apportionment using PMF while keeping computational load to a minimum. Differences in factor profiles between the seasonal PMF and time-of-day PMF runs may
indicate the effect of diurnal process changes and/or reactivity (Norris et al., 2014). Conducting PMF on smaller time windows is expected to improve results for another reason. Positive Matrix Factorization approaches have influence functions that are designed to account for the influence of outliers on the solutions (Paatero et al., 1994; Paatero et al., 1997; Ulbrich et al., 2009). These outliers depend on the time window on which the factorization is being applied (Paatero et al., 1997). A shorter time window for analysis is influenced by outliers present in that time window only, and not any other period. Thus, a shorter time
window can be expected to give higher factor resolution, given that the influence of many outliers in the dataset is removed. At the same time, the number of zeros in the dataset also assists with the quantification of PMF factors (Paatero et al., 1997). Thus, shortening time windows can also decrease the extraction of factors via PMF, as has been reported previously (Tian et al., 2017).

This paper improves upon the seasonal source apportionment previously employed in Delhi (Bhandari et al., 2020).
The Delhi Aerosol Supersite (DAS) study provides long-term chemical characterization of ambient submicron aerosol in Delhi, with near-continuous online measurements of aerosol composition (Gani et al., 2019; Arub et al., 2020; Bhandari et al., 2020; Gani et al., 2020; Patel et al., 2021a). In that study, PMF was conducted on six seasons of highly time-resolved speciated non-refractory submicron aerosol (NR-PM$_1$) organic (Org) mass spectrometer data from an Aerosol Chemical Speciation Monitor (ACSM) in the PMF receptor model at a time resolution of 5–6 min. Then, we deployed the IGOR PET tool on seasonal
datasets and 2–3 PMF factors were extracted. The extraction of a low number of factors implies low rotations, and therefore quantitative error estimation was not conducted in that study (Paatero et al., 1994).

Here, we apply the approach of conducting PMF on long-term datasets where each day was separated into six 4 hour periods with limited variability in emissions and meteorology. In our knowledge, no study has systematically assessed the use of PMF on data resolved by time of day. In this paper, we report on PMF conducted on ACSM organic aerosol data from the
two seasons of winter and monsoon 2017—collected as a part of the Delhi Aerosol Supersite (DAS) study—after resolving by time-of-day. Thus, factor MS is expected to vary in these time-of-day windows. The two seasons of winter and monsoon are selected for this analysis as they capture two extremes in seasonal concentrations, precipitation, and meteorology, especially in terms of temperature, ventilation coefficient, wind direction, and wind speed (Tables S1 and S2, Fig. S1). In addition, winter experiences extremely high organic and inorganic concentrations and high pollution episodes dominated by primary emissions
(Gani et al., 2019; Bhandari et al., 2020). We use the EPA PMF tool to apply constraints, extract a larger number of factors, and quantify errors in PMF solutions.





## 2 Methods

### 2.1 Statistical basis of approach

ME-2 is a multilinear unmixing model that can be used to perform bilinear deconvolution of a measured mass spectral matrix
(**X**) into the product of positively constrained mass spectral profiles (**F**) and their corresponding time series (**G**), as shown in
Eq. (1). In Eq. (1), **E** corresponds to the data residual not fit by the model. Given that time series and mass spectra are
deconvoluted, the model mass spectral profiles are assumed to remain constant in time. The mass balance equation underlying
the bilinear implementation of the factor analytical model and the optimization problem in the EPA PMF tool can be
represented as shown in Eq. 1–3.

$$\mathbf{X} = \mathbf{GF} + \mathbf{E} \tag{1}$$

$$x_{ij} = \sum_{p=1}^{n} g_{ip} \cdot f_{pj} + e_{ij} \tag{2}$$

Equation (2) is an elemental notation of Eq. (1). For ACSM data analysed here, $x_{ij}$ represents an element of the $m \times n$ data
matrix **X**, where $i$ represents a single time point and $j$ represents a measured ion or $m/z$. $n$ corresponds to the number of factors
in the PMF solution. Thus, $g_{ip}$ refers to the time series contribution of the $p^{th}$ factor at the $i^{th}$ time point, and $f_{pj}$ represents the
mass spectral contribution of the $j^{th}$ $m/z$ in the $p^{th}$ factor profile.

To derive factor time series and mass spectra in an iterative fitting process, ME-2 lowers residual by minimizing the
quality of fit parameter $Q$, using the gradient approach (Norris et al., 2014; Eq. (3)). Thus, PMF attempts to achieve a global
minimum to the optimization problem. $Q$ is the weighted least-squares error (sum of squares of model error normalized to
measurement error), or the summation of squares of scaled residuals of the fit at each data point. We do not expect the norm
of the actual error matrix to be zero but instead close to the ACSM measured uncertainty (an element of the measured
uncertainty is represented as $\sigma_{ij}$ in Eq. (3)). The quality of fit parameter corresponding to this uncertainty is called $Qexp$ (Ulbrich
et al., 2009). While $Qexp$ is precisely equal to $mn-p(m+n)$, for large $m$, $n$, it simplifies to $\sim mn$. Usually, PMF solutions start
from very high $Q/Qexp$, and converge to 1 as more factors are added. We refer to the $Q$ for the entire dataset as $Q0$.

$$Q0 = Min_{\mathbf{F,G}} Q = \sum_{i=1}^{m} \sum_{j=1}^{n} \left(e_{ij}/\sigma_{ij}\right)^{2} \tag{3}$$

For this discussion, we assume that Eq. (3) is subject to a constant mass spectrum $F0$, and variable time series $G0$. A
key limitation of PMF is that it assumes constant MS profiles, even though source signatures can change over the course of
the day. To address this limitation, we divide our data into six 4 hour time segments to conduct PMF analysis resolved by
time-of-day. We refer to this time-resolved organic MS-based PMF as "time-of-day PMF" and the traditional approach as
"seasonal PMF" in the paper. In the time-of-day PMF approach presented here, we minimize $Q$ separately in each of these
time-of-day windows.



## 2.2 Mathematical formulation of the time-of-day PMF approach

The mathematical formulation of the time-of-day PMF approach is introduced in Eqs. 4–17. To provide an example for splitting of data by time-of-day, we modify Eq. 3, dividing the data matrix $\mathbf{X}$ into $\mathbf{X_{day}}$ (time, $t \in$ [12 am, 12 pm)) and $\mathbf{X_{night}}$ (time, $t \in$
[12 pm, 12 am)) (Eq. 4). Here, we demonstrate that splitting the data by time-of-day will result in a better solution. Thus,

$$\mathbf{X} = \{\mathbf{X_{day}}, \mathbf{X_{night}}\} \tag{4}$$

The mathematical representation of the objective functions for conducting PMF separately for $\mathbf{X_{day}}$ and $\mathbf{X_{night}}$ periods are shown in Eq. 5 and Eq. 6 respectively. We call $Q$ for these data subsets $Q1$ and $Q2$.

$$Q1 = Min_{\mathbf{F,G}} \left( \sum_{i=1}^{m} \sum_{j=1 \ni time \in \mathbf{Xday}}^{n} (e_{ij}/\sigma_{ij})^2 \right) \tag{5}$$

$$Q2 = Min_{\mathbf{F,G}} \left( \sum_{i=1}^{m} \sum_{j=1 \ni time \in \mathbf{Xnight}}^{n} (e_{ij}/\sigma_{ij})^2 \right) \tag{6}$$

For this discussion, we assume that Eq. (5) is subject to a constant mass spectrum $\mathbf{F1}$, and variable time series $\mathbf{G1}$ for dataset $\mathbf{X_{day}}$, and Eq. (6) is subject to a constant mass spectrum $\mathbf{F2}$, and variable time series $\mathbf{G2}$ for the dataset $\mathbf{X_{night}}$. For simplification,

$$A(\mathbf{F}, \mathbf{G}) = \sum_{i=1}^{m} \sum_{j=1 \ni time \in \mathbf{Xday}}^{n} (e_{ij}/\sigma_{ij})^2 \tag{7}$$

$$B(\mathbf{F}, \mathbf{G}) = \sum_{i=1}^{m} \sum_{j=1 \ni time \in \mathbf{Xnight}}^{n} (e_{ij}/\sigma_{ij})^2 \tag{8}$$

Thus,     $$Q1(\mathbf{F1}, \mathbf{G1}) = Min(A) \text{ and } Q2(\mathbf{F2}, \mathbf{G2}) = Min(B) \tag{9}$$

Using these definitions, we can also redefine $Q0$ as shown in Eq. 10.

$$Q0(\mathbf{F0}, \mathbf{G0}) = Min(A + B) \tag{10}$$

Clearly, $Q0$ minimizes the sum of two functions $A$ and $B$. Thus, $Q0$ is a multi-objective optimization problem attempting to achieve global minimum for the combined dataset $\mathbf{X}$ (Gunantara et al., 2018; Eq. 10). The two functions $A$ and $B$ are globally
minimized separately at ($\mathbf{F1}$, $\mathbf{G1}$) in Eq. 5 and at ($\mathbf{F2}$, $\mathbf{G2}$) in Eq. 6, respectively. Thus, by definition, Eq. 5 and Eq. 6 can be written as:

$$Min_{\mathbf{F,G}} A \le A \text{ for all } (\mathbf{F}, \mathbf{G}) \tag{11}$$

$$Min_{\mathbf{F,G}} B \le B \text{ for all } (\mathbf{F}, \mathbf{G}) \tag{12}$$

Adding the inequalities in Eq. (11) and Eq. (12), we get:

$$Min_{\mathbf{F,G}}(A) + Min_{\mathbf{F,G}}(B) \le A + B \text{ for all } (\mathbf{F}, \mathbf{G}) \tag{13}$$

Since this is true for all ($\mathbf{F}$, $\mathbf{G}$), this is also true for ($\mathbf{F}$, $\mathbf{G}$) that gives the minimum of $A + B$. Thus,

$$Min_{\mathbf{F,G}}(A) + Min_{\mathbf{F,G}}(B) \le Min_{\mathbf{F,G}}(A + B), or \tag{14}$$

$$Q1 + Q2 \le Q0, or \tag{15}$$

$$Q1 + Q2 \le Q0_1 + Q0_2 \tag{16}$$

In Eq. (16), $Q0_1$ and $Q0_2$ are $Q$ contributions to $Q0$ in the ($\mathbf{F}$, $\mathbf{G}$) space corresponding to $Q1$ and $Q2$ respectively. Thus, we can see that if solutions to $Q0$ will attempt to minimize error in the ($\mathbf{F}$, $\mathbf{G}$) space corresponding to $Q1$ (minimize $Q0_1$), the obtained solution will likely worsen the error in the ($\mathbf{F}$, $\mathbf{G}$) space corresponding to $Q2$ (and therefore, not minimize $Q0_2$). This property of solutions to multi-objective optimization problems is inherent to a large class of solutions known as Pareto solutions, which





are used for source apportionment and air quality planning (Gunantara et al., 2018; Angelis et al., 2020). This limitation can
also be viewed as a limitation on the mass spectral profiles—*Q0* assumes constant mass spectral profiles for both day and night
periods, and likely fits both periods worse than the scenarios of *Q1* and *Q2*, where separate mass profiles for the two periods
were developed. Thus, in the traditional approach, varying TS on non-varying MS can only capture changes as a linear TS
scaling factor for all MS contributions. In the time-of-day PMF approach, both MS and TS are varying, and we can expect
new MS and TS patterns. For the special case of day-night data split, where equal number of points are collected in $\mathbf{X_{day}}$ and
$\mathbf{X_{night}}$, *Qexp (~ mn)* corresponding to the two matrices are equal (we call it *Qexp$_{dn}$*), whereas we call it *Qexp* corresponding to
the matrix $\mathbf{X}$ would be double that value (*2Qexp$_{dn}$*). Using these *Qexp$_{dn}$* values, Eq. 16 can be written as

$$Q1/Qexp_{dn} + Q2/Qexp_{dn} \leq Q0_1/Qexp_{dn} + Q0_2/Qexp_{dn} \qquad (17)$$

Clearly, using the day-night split thought experiment, we show that the sum of *Q1* and *Q2* (and the equivalent sum in *Q/Qexp*)
would be lower than *Q0* (and the equivalent sum of *Q/Qexp* components). By inference, dividing the time series into periods
of similar length (six 4 hour segments in this manuscript) should result in a similar relationship as Eq. (17). Overall, conducting
PMF on each of such time-of-day periods challenges the assumption of diurnally non-varying MS factors in typical PMF.

Here, we used two alternative approaches for conducting PMF. In one approach, we apply PMF by splitting the data
into six 4 hour time windows each day to illustrate the use of our time-of-day PMF method. We also conduct seasonal PMF
runs for winter and monsoon 2017 and time-of-day PMF runs for two periods (1100–1500 LT-local time and 1900–2300 LT)
in the two seasons. Thus, we conduct four time-of-day PMF runs in total. The two time-of-day periods are selected to
differentiate between influence of primary sources, changing MS due to reaction chemistry, and effect of meteorology (Table
1, Fig. S1). Results from PMF analysis for all times of the day are presented in a companion paper (Bhandari et al., 2022). In
monsoon and winter, traffic is expected to be a dominant source at night due to low cooking-related emission and overlap with
high night-time traffic on major traffic corridors (Mishra et al., 2019). At midday in monsoon, high temperatures and solar
flux imply high photochemical processing of aerosols; therefore, we expect to see more oxidized aerosols (Table 1, Fig. S1).
At winter night-time, biomass burning for heating is an expected source. We refer to the seasonal organic MS-based PMF
analysis results as "seasonal PMF" and time-of-day organic MS-based PMF analysis results as "time-of-day PMF" results in
the paper. To refer to PMF runs corresponding to specific time windows, we use the nomenclature "Season" + "Year" +
"Period" style in the format "SYYTTTT" (Table 1). For example, W171115 corresponds to the 1100–1500 LT of Winter 2017.
Using data presented in this paper, we also compare the *Q* (and *Q/Qexp*) values from the seasonal PMF runs corresponding to
the periods of the time-of-day windows (Sect. 3.4). Future work should investigate the optimal length of the time window to
sufficiently represent the diurnal variations in mass spectral profiles while managing computational burden.



Table 1 Summary of meteorology in the time-of-day PMF periods

| Season and Period | T (K) | RH (%) | VCª (m²/s) | PBLH* (m) | WS (m/s) | WD (°N) | Nomenclature |
|---|---|---|---|---|---|---|---|
| W17 1100–1500 LT | 294 | 93 | 3870 (3790) | 1353 (1356) | 2.9 | -14.0 | W171115 |
| W17 2300–0300 LT | 286 | 62 | 707 (188) | 273 (64) | 2.5 | -49.0 | W172303 |
| M17 1100–1500 LT | 308 | 82 | 6179 (6222) | 2022 (2061) | 3.1 | 6.9 | M171115 |
| M17 2300–0300 LT | 302 | 73 | 1182 (237) | 428 (84) | 2.5 | 68.0 | M172303 |

ªMedian values for VC and PBLH are reported in parenthesis.

## 2.3 Sampling site and measurements

As a part of the DAS study, an ACSM (Aerodyne Research, Billerica, MA) was operated at ~0.1 L min⁻¹ at ~1 min time resolution in a temperature-controlled laboratory on the top floor of a four-story building at IIT Delhi (Ng et al., 2011b). Additionally, BC, ultraviolet-absorbing particulate matter (UVPM), and their difference ΔC were measured using a seven-

wavelength aethalometer operated at the 1L min⁻¹ flow rate and 1 min time resolution (Magee Scientific Model AE33, Berkeley, CA) (Drinovec et al., 2015). These instruments were on separate sampling lines, both of which had a PM$_{2.5}$ cyclone followed by a water trap and a Nafion membrane diffusion dryer (Magee Scientific sample stream dryer, Berkeley, CA). Full details of sampling site, instrument setup, operating procedures, calibrations, and data processing are described in a separate publication (Gani et al., 2019).

We collected the data used in this paper in winter (January–February 2017) and monsoon (July–September 2017). Definition of the seasons comes from the Indian National Science Academy (2018) (Table 2 from Bhandari et al., 2020). Diurnal plots of meteorological variables are shown in Fig. S1. We conduct seasonal PMF runs for winter and monsoon 2017 and time-of-day PMF runs for two periods (1100–1500 LT and 1900–2300 LT) in the two seasons. We used the dataset obtained by averaging every five consecutive measurements for the seasonal PMF runs. We selected organic spectral data at a

specific set of *m/z* values between *m/z* 12 and *m/z* 120. This approach is the commonly used approach, and the reasons for the selection of the specific set of *m/z* values have been described previously (Zhang et al., 2005). Spring, summer, and autumn (mid-September to November) periods are not included in the analysis here; but seasonal PMF analysis has been presented in previous publications (Bhandari et al., 2020; Patel et al., 2021a).

## 2.4 PMF tool and runs

The EPA PMF v5.0 was used to conduct ME-2 analysis on this dataset and interpret its results (Norris et al., 2014). Further details on the statistical basis of this method are available elsewhere (Paatero et al., 1999; Paatero et al., 2002). For the base run, the iterative PMF technique does not make any assumptions for source or time profiles. If factors extracted in the base run were not clearly associated with a source type but suggestive of the presence or mixing of specific sources, constraints were applied on the factors in the base run to extract cleaner source profiles (Brown et al., 2012; Brown et al., 2015). An R





package was developed to automate the process of data analysis of EPA PMF outputs (R Core Team, 2019). We readjusted the results from PMF analysis to account for underestimation of factor mass based on the selected *m/z* values only. To account for particle losses, we applied transmission and collection efficiencies after conducting PMF analysis (Gani et al., 2019). Additional details of the steps for conducting PMF, R code, and criteria for factor selection have been discussed in detail in the Supplement (Sect. S1).

**2.5 Uncertainty estimation**

In EPA PMF, quantitative error estimation (EE) of random error and rotational ambiguity was conducted using Bootstrapping (BS), Displacement (DISP), and Bootstrapping enhanced with Displacement (BS-DISP). The algorithms of these EE techniques are described in detail elsewhere (Paatero et al., 2014). The application of these EE techniques leads to several orders of magnitude increase of computational time and memory requirements in conducting PMF runs (Paatero et al., 2014).

Bootstrapping or BS estimates "disproportionate effects of a small set of observations on the solution". In the process, BS accounts for random error and to a limited extent rotational ambiguity (Norris et al., 2014). EPA PMF automatically identifies BS datasets using the parameter "Block Size" that is based on the principle of stationarity and accounts for underlying serial correlations (Politis and White, 2004). The default calculation of the "Block size" in EPA PMF is based on incorrect calculations and updated calculations have been published (Patton et al., 2009) but not implemented in the EPA PMF tool. We

used the corrected block size estimation procedure as shown in the Supplement (Sect. S2, Table S5). BS factors are then mapped to base factors using the parameter "Minimum Correlation R-Value", which is the minimum Pearson correlation coefficient used for BS factor assignment. We use the default value of 0.6. We conduct 100 BS runs for each PMF solution. Specification of too many factors in the base model may create artificial PMF factors (Ulbrich et al., 2009). BS factors with rotational ambiguity may also get mapped to other base factors. This scenario is called factor swapping and occurs for not-

well-defined (NWD) solutions. These factors will likely have low BS mapping with their equivalent base run factors (Paatero et al., 2014). We only finalize PMF factor solutions with approximately 80% or more BS mapping for all PMF factors.

    Displacement or DISP estimates rotational ambiguity in PMF solutions by identifying the range of allowable MS profile contributions in the PMF factors. Bootstrapping enhanced with Displacement or BS-DISP combines the bootstrap and displacement techniques to simultaneously estimate random error and rotational ambiguity in PMF solutions. In BS-DISP, BS

resamples explore the solution space randomly and DISP explores the rotationally accessible space around each BS resample. The ranges in DISP are obtained corresponding to four limits on changes in the *Q*-value (*dQ*-max): 4, 8, 15, and 25. BS-DISP also reports ranges for contributions at different *m/zs* to MS profiles of PMF factors. These ranges correspond to four limits on changes in the *Q*-value (*dQ*-max): 0.5, 1, 2, and 4. The obtained PMF factors using both approaches are then mapped to base factors, and the number of cases of factor swaps are noted. Sometimes, DISP and BS-DISP runs are terminated when

encountering large changes in the *Q*-value, which suggests the base case solution is not close to the global minimum. Generally, small changes in *Q* suggest PMF solutions are close to the global minimum. Additionally, small number of factor swaps suggest low rotational ambiguity and robustness of the PMF solution. We only finalize PMF solutions with very few swaps at





the smallest $dQ$-max value. Some DISP and BS-DISP runs terminated due to computational limits or encountering high $dQ$-max. For these cases, we used the number of factor swaps at termination as an estimate of total factor swaps. Finally, even

when solutions with factor swaps are encountered, only solutions with swaps among the lowest number of factors are considered interpretable (Norris et al., 2014). All other solutions are rejected.

## 3 Results and discussion

In this paper, we focus on the implementation of the time-of-day PMF technique on organic aerosol measured during monsoon midday and night periods and winter midday and night periods (Table 1). We report average concentrations of PMF factors in

Table 2. For reference, data from seasonal PMF analysis are also presented. We find that time-of-day PMF analysis: (i) generates a larger diversity of primary factors than seasonal PMF, (ii) resolves mass spectra of cooking-related factors such as cooking organic aerosol (COA), mixed COA-HOA, and solid fuel combustion organic aerosol (SFC-OA) in Delhi, that are relatively unexplored (Tobler et al., 2020), and (iii) resolves different kinds of biomass burning organic aerosol (BBOA)-related factors (two BBOAs, one SFC-OA) based on MS and TS correlations (Sect. 3.3) (Table 2). Seasonal monsoon PMF

analysis represents primary organic aerosol (POA) by a single hydrocarbon-like organic aerosol (HOA) whereas monsoon time-of-day PMF analysis represents midday POA as a mixed COA-HOA factor and night-time POA as separate HOA and COA. In winter, seasonal PMF analysis separates POA into HOA and BBOA factors. In winter time-of-day PMF analysis, midday POA separates into an SFC-OA factor and a BBOA factor, and night-time PMF analysis gives HOA and BBOA. All analyses generate two oxidized organic aerosol (OOA) factors. Time series of the different time-of-day PMF factors are shown

in Figs. 1–2.

In Sect. 3.1, we discuss the mass spectral profiles (MS) and time series patterns (TS) of factors obtained in seasonal PMF analysis conducted for winter and monsoon. In Sect. 3.2, we discuss the mass spectral profiles and time series patterns of factors obtained in time-of-day PMF analysis conducted for winter and monsoon midday and night-time periods. In Sect. 3.3, we contrast the mass spectra and time series patterns of primary and secondary PMF factors obtained from time-of-day

and seasonal PMF analyses. The mass spectra of POA, a proxy for primary OA, and OOA, a proxy for secondary OA were calculated by adding the component factors corresponding to each type (e.g., POA = HOA + BBOA + COA) weighted by their respective time series contributions. This estimation allows a comparison between the results from the time-of-day and seasonal analyses. In Sect. 3.4, we discuss period-specific $Q$ (and $Q/Qexp$) values for the time-of-day PMF approach and the seasonal PMF approach. We also compare the $Q/Qexp$ TS patterns and $Q/Qexp$ by $m/z$ to identify periods and $m/z$s with

particularly significant changes in $Q/Qexp$.



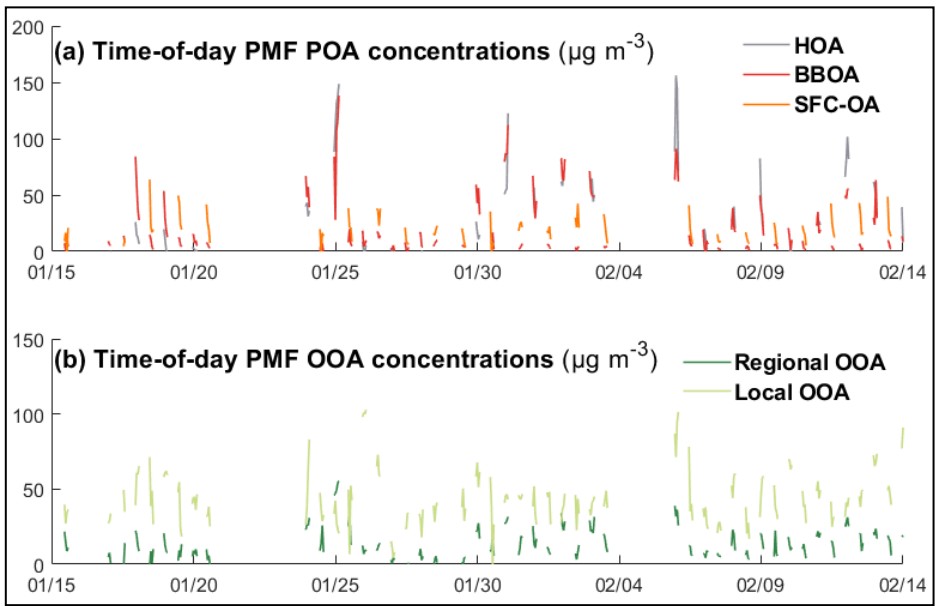

**Figure 1 shows hourly averaged seasonally representative concentration time series of time-of-day PMF (a) primary and (b) secondary factors for winter 2017 (in μg m⁻³). Chopped lines are representative of the time-of-day PMF sub-setting. POA PMF factors show stronger variability than OOA PMF factors.**

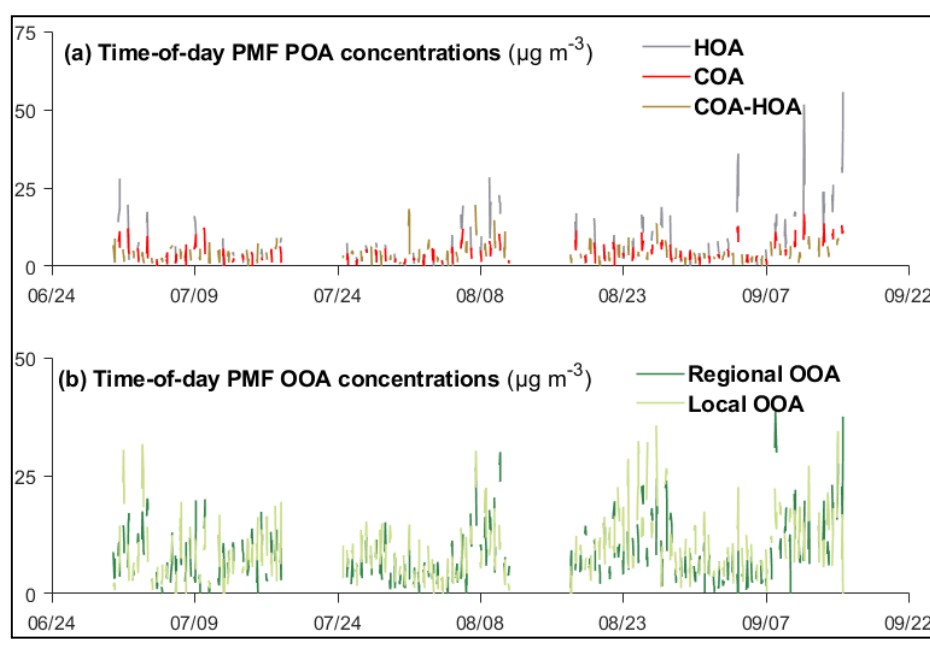


**Figure 2 shows hourly averaged seasonally representative concentration time series of time-of-day PMF (a) primary and (b) secondary factors for monsoon 2017 (in μg m⁻³). Chopped figures are representative of the time-of-day PMF sub-setting. POA PMF factors show stronger variability than OOA PMF factors.**






**Table 2 PMF factor concentrations in seasonal PMF and time-of-day PMF analysis (in µgm⁻³).**

| Season | Period | PMF Run Type | Factor Type | Factor Concentrations |
|---|---|---|---|---|
| M17 | 1115 | Seasonal | POA (HOA) | 2.5 (2.5) |
| | | | OOA (Local OOA, Regional OOA) | 18.7 (5.4, 13.3) |
| | | Time resolved | POA (COA-HOA) | 4.0 (4.0) |
| | | | OOA (Local OOA, Regional OOA) | 17.4 (6.4, 11.0) |
| | 2303 | Seasonal | POA (HOA) | 8.8 (8.8) |
| | | | OOA (Local OOA, Regional OOA) | 21.4 (10.7, 10.8) |
| | | Time resolved | POA (HOA, COA) | 12.1 (7.7, 4.4) |
| | | | OOA (Local OOA, Regional OOA) | 17.8 (7.7, 10.0) |
| W17 | 1115 | Seasonal | POA (HOA, BBOA) | 13.3 (3.5, 9.7) |
| | | | OOA (Local OOA, Regional OOA) | 55.5 (13.0, 42.4) |
| | | Time resolved | POA (SFC-OA, BBOA) | 23.0 (18.1, 4.9) |
| | | | OOA (Local OOA, Regional OOA) | 46.6 (37.6, 8.9) |
| | 2303 | Seasonal | POA (HOA, BBOA) | 86.3 (49.3, 37) |
| | | | OOA (Local OOA, Regional OOA) | 56.5 (22.3, 34.2) |
| | | Time resolved | POA (HOA, BBOA) | 71.8 (35.5, 36.2) |
| | | | OOA (Local OOA, Regional OOA) | 70.9 (18.7, 52.2) |

### 3.1 Seasonal PMF runs

The analysis in this section focuses on the PMF factors from seasonal PMF analysis; since this work focuses on specific times
of day, the results are presented only for the 1115 and the 2303 time windows. Due to differing meteorology, sources, and
photochemistry, the factor speciation, their mass spectra, and their time series patterns are quite different in the two seasons.
A comparison of POA and OOA in different seasons has been previously presented (Bhandari et al., 2020). In winter, seasonal
PMF analysis results in two factors representing POA, namely HOA and BBOA, whereas only HOA is obtained in monsoon
seasonal PMF analysis. In the two seasonal PMF runs, we also obtain two OOA factors: local (less oxidized) OOA and regional
(more oxidized) OOA (Drosatou et al., 2019; Table 2).

The behaviour of the HOA factor MS is in line with the reference HOA factor MS, as suggested by the dominance of
hydrocarbon signatures in the HOA spectrum belonging to the series $C_nH_{2n-1+}$ and $C_nH_{2n+1+}$ (Ng et al., 2011a; Bhandari et al.,
2020; Pearson R~0.95; Figs. S4 and S5). In monsoon, however, the seasonal PMF HOA MS is also strongly correlated with
the reference COA factor MS (Ng et al., 2011a; Pearson R~0.90; Fig. S5). In winter, the fractional contributions of the BBOA
factor MS at $m/z$s 60, 73, and 115 are in line with the reference BBOA factor MS (He et al., 2010; Crippa et al., 2014; Bertrand





et al., 2017; Pearson R~0.90; Fig. S6). As expected, POA tracers, carbon monoxide (CO) and black carbon (BC), correlate more strongly with HOA and BBOA factor TS than with the OOA factor TS (Figs. S7 and S8). Additionally, BBOA correlates with chloride, particularly in the evening, suggestive of agricultural and other open waste burning-related contributions (Li et al., 2014a, b; Kumar et al., 2015; Fourtziou et al., 2017; Spearman R~0.70; Figs. S7 and S9). We also observe strong
correlations of the local OOA factor with chloride (Spearman R~0.65; Fig. S7). These results are consistent with our previous seasonal organic-inorganic PMF analysis which suggested that chloride, associated with an oxidized BBOA factor (likely a combination of local OOA and BBOA) with weak $BC_{BB}$ and $\Delta C$ correlations, might be linked to an industrial source (Bhandari et al., 2020). Indeed, chloride has weak correlations with $BC_{BB}$ (Spearman R~0.45; Figs. S7 and S9).

OOA factors are principally associated with secondary organic aerosol (SOA; Zhang et al., 2011). Mass spectra of
both local OOA and regional OOA correlate strongly with the reference OOA factor (Figs. S10–S11a–b; Pearson R≥0.95). However, local OOA correlates more strongly with the reference semi-volatile oxidized organic aerosol (SVOOA) factor (Zhang et al., 2011; Drosatou et al., 2019; Figs. S10 and S11; Pearson R~0.80). The time series of the regional OOA factor correlates stronger with sulfate, whereas local OOA correlates stronger with chloride and black (BC) and brown carbon (UVPM) (Figs. S7 and S8). Overall, regional OOA shows less diurnal variability than local OOA, in line with a regional origin
(Figs. S12 and S13). Detailed 15 min time series patterns of seasonal PMF factors for the midday (1115) and nighttime (2303) periods in the two seasons are discussed in the Supplement (Sect. S3).

### 3.2 Time-of-day PMF runs

The analysis in this section focuses on the PMF factors from time-of-day analysis for the 1115 and the 2303 time windows. Here, we show that time-of-day PMF analysis resolves mass spectra of cooking- and biomass burning-related factors (one
COA, one mixed COA-HOA, one SFC-OA, two BBOAs) based on MS and TS correlations. Only nighttime periods separate clean HOA factors.

### 3.2.1 Primary factor MS and TS

#### Winter 2017 Primary factor MS

At winter midday, PMF analysis results in two factors representing POA, SFC-OA and BBOA, whereas at nighttime, HOA
and BBOA are obtained (Table 2). The behavior of the winter time-of-day PMF HOA factor MS is in line with the reference HOA factor MS (Ng et al., 2011a; Pearson R>0.95; Fig. S17a). The MS of the winter BBOA factors obtained are correlated with the reference profile but differ in contributions at key *m/z* values such as *m/z*s 29, 43, and 44 (Pearson R≥~0.8; Fig. S16b, S17b). Both MS profiles show much larger *m/z* 29 contributions than the reference profile, suggesting a strong influence of wood burning (Bahreini et al., 2005; Schneider et al., 2006). The winter midday BBOA is more oxidized (MS shows higher
ratio of contributions at *m/z* 44 to *m/z* 43) and shows a low *m/z* 60 contribution. It also has a high contribution at *m/z* 15 (Fig. S16b). Similar BBOA MS profiles with high *m/z* 15 have been observed previously as well (Crippa et al., 2013). In contrast, the winter nighttime BBOA is less oxidized and BBOA MS shows an *m/z* 60 contribution closer to the higher end of the





reference profile (Fig. S17b). At midday in winter, we also obtain a mixed POA factor (Fig. S16a). We call it solid fuel combustion organic aerosol (SFC-OA) as the factor MS correlates with multiple reference MS profiles such as BBOA, HOA,

and COA (Pearson R>0.8). This behavior is similar to a seasonal PMF SFC-OA factor identified recently in ToF-ACSM analysis for NR-PM$_{2.5}$ in Delhi. In that study, that factor was expected to be associated with heating- and cooking-related domestic fuel combustion and open-fire activities (Tobler et al., 2020; correlation at all *m/z*s but *m/z* 44, Pearson R>0.95; Fig. S18).

Monsoon 2017 Primary Factor MS

At midday in monsoon, we see only one POA factor, COA-HOA (Fig. S19). At monsoon nighttime, HOA and COA separate out (Fig. S20a–b). The behavior of the monsoon nighttime time-of-day PMF HOA factor MS is in line with the reference HOA factor MS (Ng et al., 2011a; Pearson R>0.95; Fig. S20a). The monsoon nighttime COA factor MS is very similar to the reference COA factor MS (Pearson R~0.90; Robinson et al., 2018, ratio of contributions at *m/z* 55 to *m/z* 57~1.66; Fig. S20b). A key feature of this COA factor is the high *m/z* 41, a characteristic feature of COA from heated cooking oils, especially in

Asian cooking (Allan et al., 2010; Liu et al., 2018; Zhang et al., 2020; Zheng et al., 2020). At midday in monsoon, we observe a mixed COA-HOA factor, and COA-HOA MS shows similarities with both the reference COA and HOA MS (ref. COA: Pearson R~0.90, ref. HOA: Pearson R~0.80; Fig. S19). The inability to separate HOA and COA factors for mass spectral data obtained in a major city in the Indo-Gangetic Plains has been observed previously as well (Thamban et al., 2017; Bhandari et al., 2020). However, a key difference of this factor compared to the reference HOA and COA profiles are the large contributions

at *m/z* 44 in monsoon midday COA-HOA. These high contributions are likely a result of the highly oxidizing environment at afternoon. Afternoon overlaps with periods of high shortwave radiative flux (SWR) and therefore high reactivity of the atmosphere (Fig. S1).

Primary Factor TS

CO and BC serve as tracers for HOA, BBOA, and SFC-OA (Figs. S21–S24). However, SFC-OA also has the strongest correlations among all PMF factors with nitrate and chloride (Fig. S21). Like winter midday SFC-OA, the monsoon midday COA-HOA factor also has the strongest correlations among all factors with chloride (Fig. S23). A correlation with chloride is suggestive of the influence of landfill emissions, trash burning, and solid-fuel sources (Dall'Osto et al., 2015, Lin et al., 2017). Otherwise, COA-HOA has weak correlations with external tracers. Similar behavior of COA-dominated factors has been seen

previously as well (Huang et al., 2010, Sun et al., 2011, Liu et al., 2012, Sun et al., 2013, Hu et al., 2016, Stavroulas et al., 2019). Among the two BBOA obtained, winter midday BBOA correlates strongly with chloride (Fig. S21). At nighttime however, winter BBOA correlates strongest with the wood burning component of BC (BC$_{BB}$) and weakly with chloride, suggesting at least two different origins of BBOA (Fig. S22). This is consistent with our previous work, where we have separated BBOA-like factors with different correlations with chloride and BC$_{BB}$ in different seasons (Bhandari et al., 2020;

Patel et al., 2021a).



### 3.2.2 Secondary factor MS and TS

Time-of-day PMF and seasonal PMF generate two OOA factors, local OOA and regional OOA, in each run (Figs. S25 and S26). The time-of-day PMF OOA factors show MS and TS behaviour similar to the seasonal PMF OOA factors, as shown in Sect. 3.3. Mass spectra of both local OOA and regional OOA correlate strongly with the reference OOA factor (Pearson R>~0.80). In all cases, the regional OOA factor MS shows higher correlations with the reference OOA MS and/or lower correlations with the reference SVOOA MS compared to the corresponding local OOA MS (Figs. S25 and S26). Also, we consistently observe that the more oxidized regional OOA factors have flatter diurnal time series patterns than the less oxidized local OOA factors, in line with their expected average lower volatility and contributions from long-range transport (Drosatou et al., 2019; Figs. S27–S30). We see an overlap of external tracers suggesting mixing of the two OOA components (see Sect. S4). This is not surprising since the oxidized components are present in the atmosphere for substantial time and could have undergone mixing (Drosatou et al., 2019). While combined organic-inorganic PMF yields a clear separation of anion-associated OOA components, such PMF analysis is beyond the scope of the current publication (Bhandari et al., 2020). Since we observe factor mixing of the two secondary components, detailed analysis of the factor MS and TS (correlations with external tracers, features of the mass spectra) are presented in the Supplement (see Sect. S4).

### 3.2.3 Time series patterns of time-of-day PMF factors

Time series patterns exhibit contrasting behaviour in winter and monsoon time-of-day PMF analysis, similar to the seasonal factor contrast (Sect. S3; Figs. 3–4a–b). Midday, concentrations of all primary factors exhibit a monotonically decreasing pattern likely due to increasing ventilation (Figs. 3a, 4a, S1). In the midday period, winter peak SFC-OA and BBOA concentrations are both ~3 times the period minimum (Fig. 3a). At winter night, peak concentrations of HOA and BBOA are ~2.5 times and ~3 times the period minimum (Fig. 3b). In contrast, monsoon primary factors exhibit lower variability midday (peak COA-HOA concentrations~2 times the period minimum) and night-time (peak HOA~2.5 times the period minimum; peak COA~2 times the period minimum).

Additionally, the night-time factors in both seasons show larger differences between the mean and the median than the corresponding midday factors in the same seasons, which suggests episodic nature of factors. The presence of episodes in these primary factors could be a consequence of the temperature-related inversions at night-time, which lead to aerosol accumulation (Bhandari et al., 2020). These episodes could also be a result of episodic sources contributing to these factors. Generally, HOA shows the largest mean-median differences, and episodic contributions could be from heavy duty vehicles, brick kilns, and construction and road paving activities (Guttikunda et al., 2013; Dallmann et al., 2014; Mishra et al., 2019; Khare et al., 2020; Misra et al., 2020). For BBOA, these sources could be associated with burning events, as hypothesized previously (Bhandari et al., 2020). Episodic events could also be due to precipitation (Fig. S1). OOA factors experience mixing, so their time series patterns are not discussed.



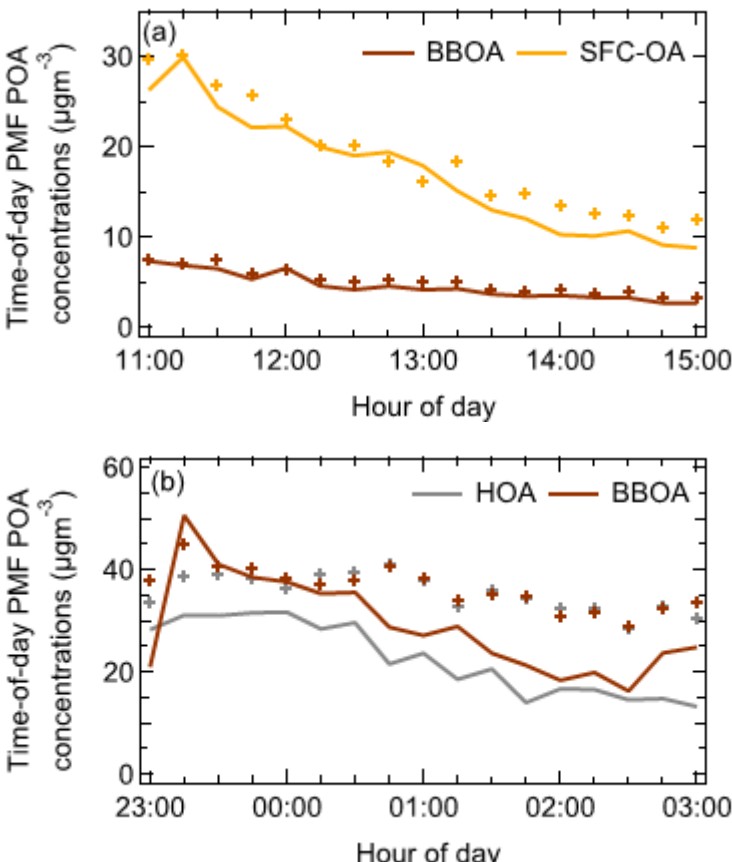

**Figure 3 shows 15 min averaged seasonally representative mean (+) and median concentrations (lines) of time-of-day PMF primary**
**factors for the periods: (a) W171115 and (b) W172303 (in µg m⁻³). Night-time factors show evidence of episodes.**

### 3.3 Comparisons of POA and OOA MS and TS obtained using time-of-day PMF and seasonal PMF

Results from the previous sections show that time-of-day PMF analysis generates a larger diversity of factors compared to
seasonal PMF analysis. In this section, we summarize and compare the primary and secondary MS and TS contributions of
factors using PMF results from two approaches—seasonal PMF and time-of-day PMF analysis. We show that (i) seasonal
PMF analysis significantly underestimates primary concentrations at midday compared to time-of-day analysis (Tables 2 and
3; Figs. 5–6a–b), (ii) midday shows cleaner signatures in POA factor MS in time-of-day PMF analysis compared to the seasonal
PMF analysis (Figs. 7–8a–b), and (iii) night-time OOA MS and TS show larger differences between the two techniques than
night-time POA MS and TS, whereas midday shows larger differences in POA than OOA (Table 3, Figs. 7–8a–b, S31–S36).
We also observe larger differences of the POA mass spectra from the two techniques midday than at night (Figs. 7–8a–b).
Detailed MS and TS comparisons of time-of-day PMF POA and OOA with the seasonal PMF results are discussed below.

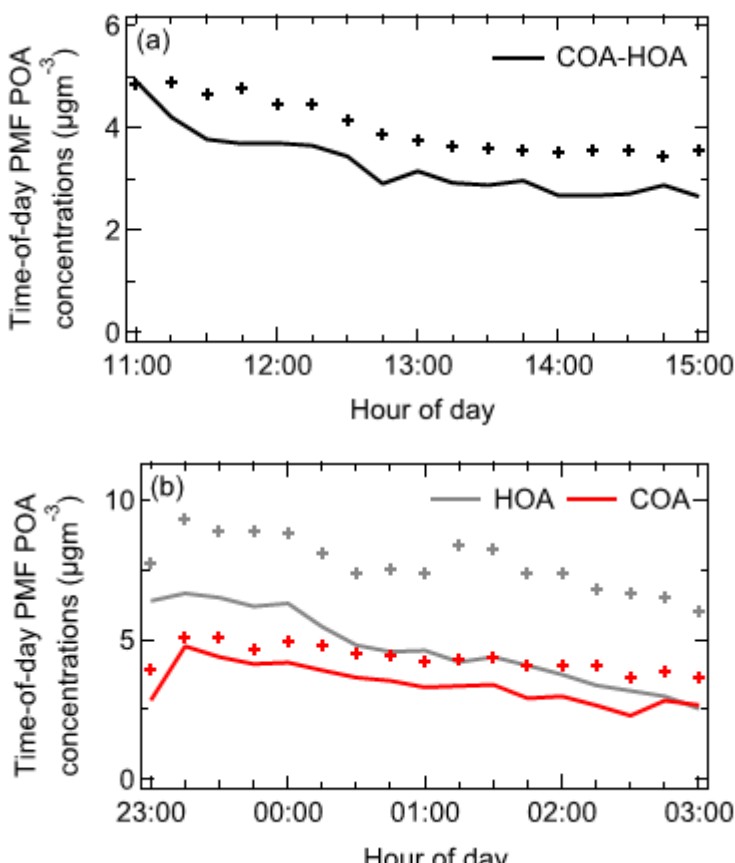

**Figure 4 shows 15 min averaged seasonally representative mean (+) and median concentrations (lines) of time-of-day PMF primary factors for the periods: (a) M171115 and (b) M172303 (in µg m⁻³). Night-time HOA shows stronger episodes than COA.**

**Table 3 Time series correlations of time-of-day POA and OOA TS with seasonal POA and OOA TS**

| Period | Factor Type | Pearson R | Slope/intercept with the corresponding seasonal POA/OOA TS |
|---|---|---|---|
| W171115 | POA | 0.96 | 1.38/4.7 |
| | OOA | 0.96 | 0.77/3.9 |
| W172303 | POA | 1.00 | 0.94/-9.2 |
| | OOA | 0.98 | 1.19/3.4 |
| M171115 | POA | 0.86 | 1.04/1.4 |
| | OOA | 0.99 | 0.99/-1.2 |
| M172303 | POA | 0.99 | 1.12/2.3 |
| | OOA | 0.99 | 0.86/-0.6 |

Here, we show that the time-of-day PMF approach shows strong similarities in time series patterns of primary factors compared to seasonal PMF analysis. However, the two approaches show substantial time-of-day dependent differences in detected mass spectra.



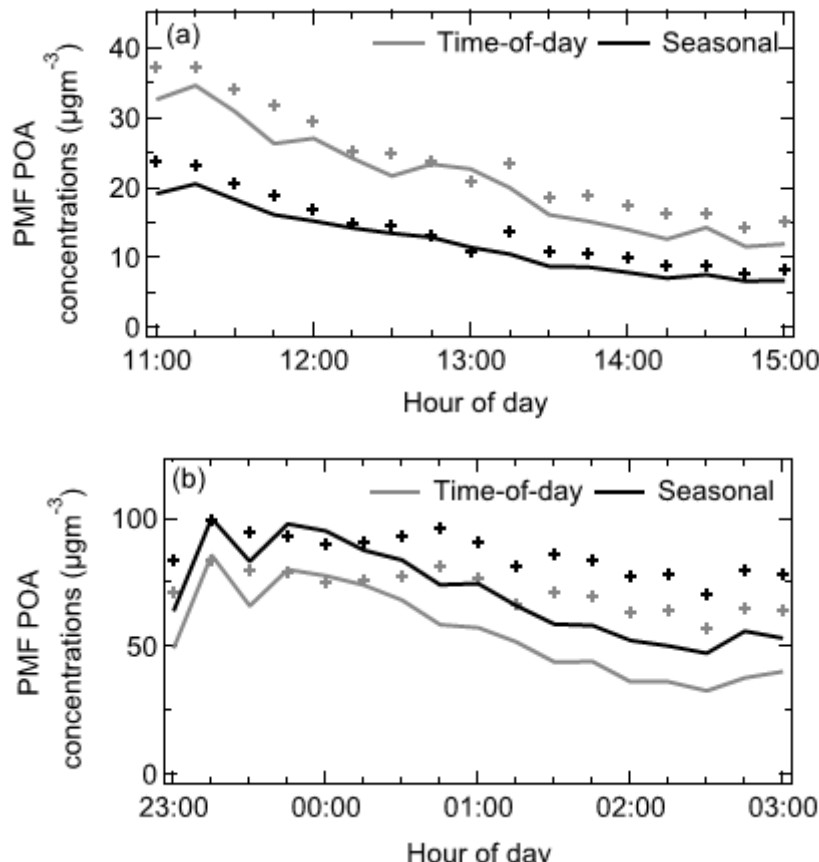


**Figure 5 shows 15 min averaged seasonally representative diurnal mean (+) and median (lines) concentration time series of POA for the periods: (a) W171115 and (b) W172303 (in μg m$^{-3}$). Night-time factors show evidence of episodes.**

### 3.3.1 Comparison of POA time series

The behaviour of POA is consistent with the individual component primary factors. POA is monotonically decreasing in all
periods, concentrations are more variable during the day, and night-time concentrations are several times those of midday concentrations (Figs. 5–6a–b). We observe striking similarities of the 15 min averaged time series patterns of POA between the two techniques across all periods (Table 3, Pearson R>0.85). The strong linear correlations suggest that time-of-day PMF analysis results in shifted (but correlated) TS patterns.

### 3.3.2 Comparison of POA mass spectra

The time-of-day PMF approach generates POA mass spectra both similar and different from the seasonal PMF approach, depending on the time-of-day (Figs. 7–8a–b). Two features stand out in these comparisons: the midday POA MS are dissimilar at key $m/z$s whereas the night-time POA MS are nearly identical.



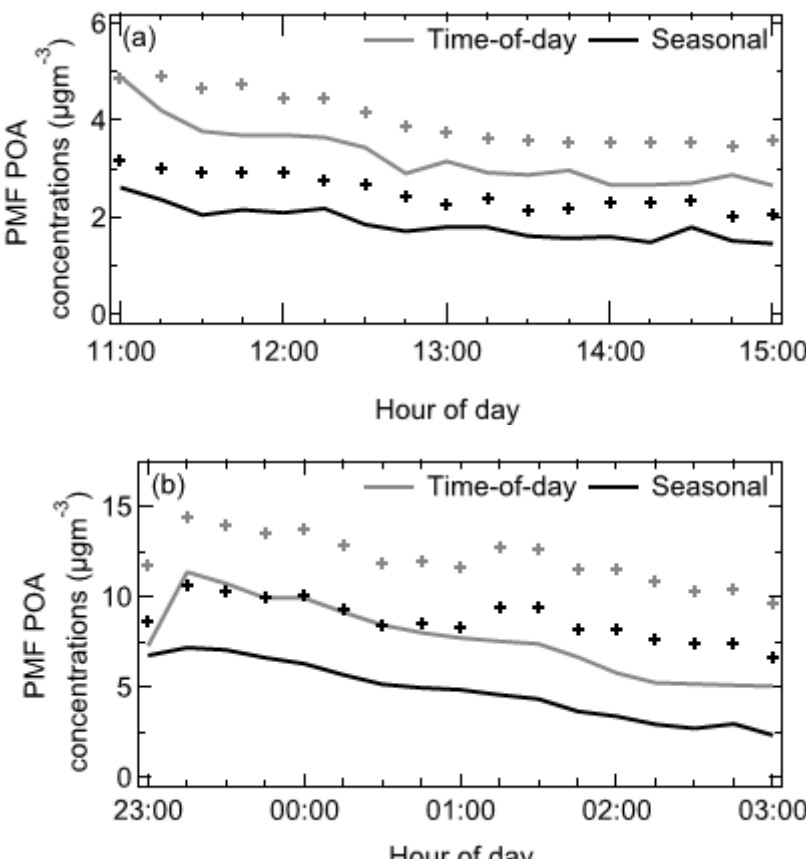

**Figure 6 shows 15 min averaged seasonally representative diurnal mean (+) and median (lines) concentration time series of POA for the periods: (a) M171115 and (b) M172303 (in μg m⁻³). Night-time factors show stronger evidence of episodes.**

Figures 7–8a–b show the MS pattern of time-of-day PMF POA and seasonal PMF POA at winter midday and night-time (Fig. 7a–b) and monsoon midday and night-time (Fig. 8a–b). In time-of-day PMF POA presented here, we observe a lower ratio of contributions at $m/z$ 43 to $m/z$ 44 than seasonal PMF POA. This lower ratio is indicative of the more oxidized nature of the POA factor compared to the seasonal POA (Ng et al., 2010). At midday, we observe higher contributions in time-of-day PMF POA at $m/z$ 44; in line with the high photochemical processing (SWR flux, Fig. S1). We also observe a higher ratio of contributions at $m/z$ 55 to $m/z$ 57, and lower contributions at $m/z$ 57 (Figs. 7a and 8a). These observations are in line with a strong cooking influence (and lower traffic influence) at midday (Ng et al., 2011a; Robinson et al., 2018). At winter midday, we also observe lower contributions at $m/z$s 29, 60, and 73 in time-of-day PMF than seasonal PMF (Bahreini et al., 2005; Schneider et al., 2006). This observation is likely a consequence of the removal of the influence of wood burning for night-time space heating on the time-of-day PMF POA MS for midday. At monsoon midday, we observe a higher contribution at $m/z$ 41 than $m/z$ 43 in time-of-day PMF POA, which is indicative of the influence of cooking (Allan et al., 2010; He et al., 2010). This POA also shows higher contribution at $m/z$ 29, suggesting a higher influence of wood burning, likely associated with midday cooking. At night-time, the differences between the time-of-day and seasonal profiles are much smaller. Overall,





time-of-day PMF analysis seems to be capturing very specific features of primary aerosol behaviour better than seasonal PMF analysis.

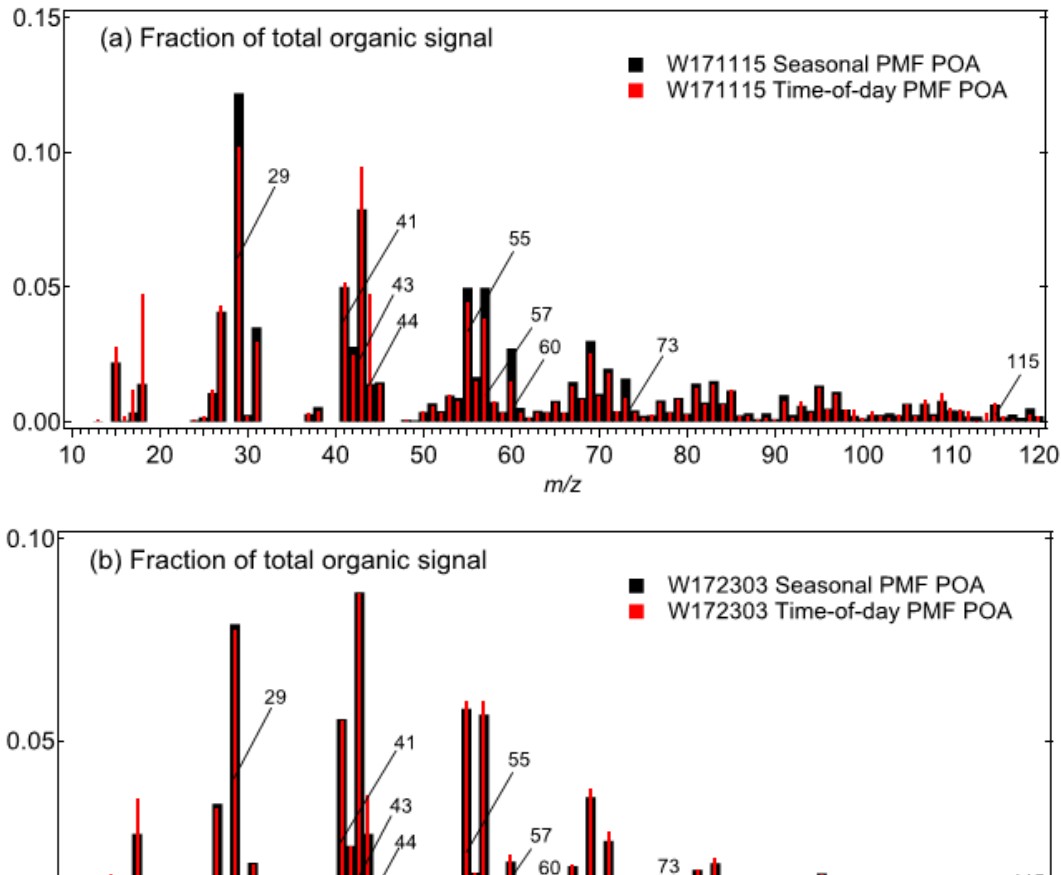

**Figure 7 shows the mass spectra of time-of-day PMF POA and seasonal PMF POA for the periods (a) midday and (b) night-time in winter 2017. Midday MS shows larger differences compared to night-time MS.**





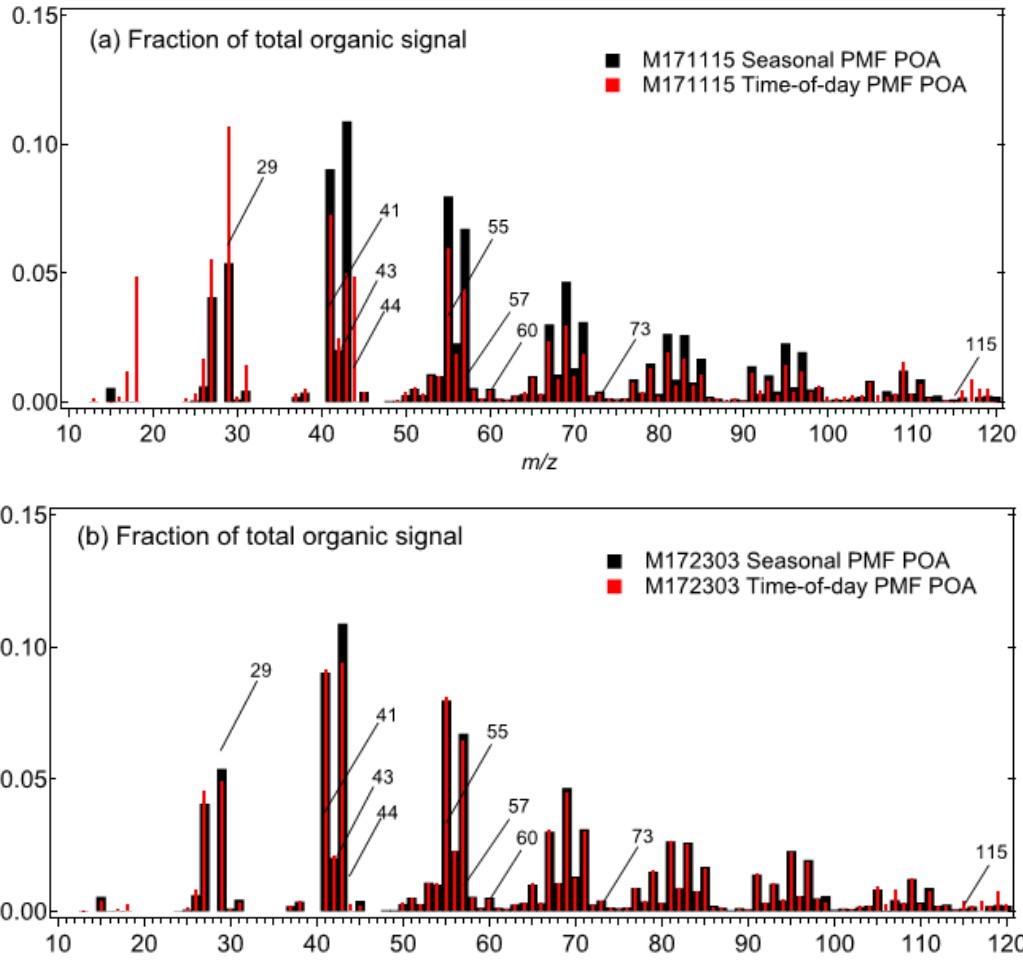


**Figure 8 shows the mass spectra of time-of-day PMF POA and seasonal PMF POA for the periods (a) midday and (b) night-time in monsoon 2017. Midday MS shows larger differences compared to night-time MS.**

We can also compare midday and night-time POA MS from the time-of-day PMF analysis and the seasonal PMF analysis (Figs. S31–S32a–b). For both seasons, the two comparisons (seasonal PMF and time-of-day PMF) of midday and

night-time POA MS indicates more primary nature at night-time than midday, based on the higher contributions at the $m/z$s corresponding to the alkyl hydrocarbons associated with primary combustion (Zhang et al., 2011). However, this contrast is sharper in time-of-day PMF analysis, in line with the ability of the approach to capture variable MS (Figs. S31b and S32b). The seasonal PMF midday–night-time comparison also fails to capture the influence of cooking midday based on the low ratio of contributions at $m/z$ 55 to $m/z$ 57 (Figs. S31a and S32a). This contrast between midday and night-time is particularly high

in monsoon time-of-day PMF analysis (Fig. S32b). While seasonal PMF analysis for monsoon suggests no change in MS between midday and night-time, time-of-day PMF analysis suggests large shifts in contributions at key $m/z$s such as 41, 43,





44, 55, and 57, in line with the changing importance of cooking from midday to night. These differences demonstrate the ability of time-of-day PMF to capture variable MS.

We can also compare OOA MS and TS as well as conduct midday and night-time comparisons for time-of-day PMF
and seasonal PMF analysis (Sect. S4). Time-of-day PMF approach shows very strong similarities in OOA time series patterns and similar MS compared to the seasonal PMF analysis. Also, comparisons of OOA TS patterns exhibit several similarities to the POA TS comparisons. Like POA TS, OOA TS concentrations are more variable during the day, and 15 min averaged time series patterns of OOA between the two techniques are very similar across all periods (Figs. S33–S34a–b, Table 3, Pearson R>0.95). The time-of-day PMF approach generates OOA mass spectra broadly similar to the seasonal PMF approach, with
major differences only at *m/z* 44 (Figs. S35–S36a–b). Comparisons of midday and night-time time-of-day PMF OOA MS, shows interesting patterns not apparent in seasonal PMF analysis (Figs. S37–S38a–b). For example, time-of-day PMF analysis for monsoon 2017 suggests lesser oxidized OOA at midday than night-time, likely caused by the presence of semi-volatile compounds (Fig. S38b). Similar behaviour has been observed elsewhere as well, and was attributed to biogenic emissions (Canonaco et al., 2015).

**3.4 Quantification of quality of fit using *Q* and *Q/Qexp* patterns**

As discussed in the methods section, PMF iterates to identify minima in the *Q* value, a residual-based metric often used as a measure of the quality of fit of the PMF solution (Sect. 2.2). Here, we compare the time-of-day PMF and the seasonal PMF approaches based on their *Q* and *Q/Qexp* patterns. We show that *Q* and *Q/Qexp* are lower in time-of-day PMF analysis than seasonal PMF analysis. By allowing the MS to change substantially relative to the seasonal profile at specific times of day, the
time-of-day PMF lowers the residuals and therefore the *Q* values. These improvements in *Q* are (i) larger in winter compared to monsoon, (ii) larger at midday than night-time, and (ii) are non-monotonic within the time-of-day periods.

**3.4.1 Comparison of average *Q* and *Q/Qexp* in different time-of-day periods**

In Table 4, we compare the average *Q* and *Q/Qexp* values obtained in the time-of-day PMF analysis and the seasonal PMF results. Our results indicate that time-of-day PMF approach significantly improves *Q* by 6–55% and *Q/Qexp* by 5–30% of the
original *Q* and *Q/Qexp* values, respectively. A part of the improvement in *Q* going from seasonal PMF to time-of-day PMF is also due to the lower number of points and therefore, lower degree of freedom, as well as larger number of weak *m/zs* (Paatero et al., 1994; Paatero et al., 1997; Ulbrich et al., 2009; Table S3). However, decreases occurring in *Q/Qexp* are less affected by the different number of weak *m/zs*, and validate the improvement (Table 4). Winter midday observes larger seasonal *Q* and *Q/Qexp* values than monsoon midday despite lower number of time series points at winter midday. This result is likely an
effect of the larger diversity of sources expected in winter, and a limitation of seasonal PMF to capture sources through static MS profiles (Paatero et al., 2002). Drops in monsoon and winter midday *Q/Qexp* (going from seasonal PMF to time-of-day PMF) are likely an outcome of the factor switching from only HOA to cooking-related factors (COA-HOA and SFC-OA, respectively). Further, even though seasonal *Q/Qexp* at winter night-time is higher than monsoon night-time, time-of-day





*Q/Qexp* is similar. Improvements at night-time comes primarily from a change in the OOA MS, as shown in Sect. 3.3. Thus,
time-of-day PMF results in large improvements in fit relative to the seasonal PMF analysis.

**Table 4 Comparison of average *Q* and *Q/Qexp* in time-of-day PMF and seasonal PMF**

| Period | Seasonal PMF *Q*[a] | Time-of-day PMF *Q* | Seasonal PMF *Q/Qexp*[a] | Time-of-day PMF *Q/Qexp* | % Change *Q* | *Q/Qexp* |
|---|---|---|---|---|---|---|
| M171115 | 288030 | 241858 | 1.84 | 1.74 | -16% | -5% |
| M172303 | 333134 | 313170 | 2.16 | 1.93 | -6% | -11% |
| W171115 | 369452 | 164975 | 4.36 | 3.05 | -55% | -30% |
| W172303 | 197984 | 161468 | 2.37 | 1.95 | -18% | -18% |

[a]The seasonal PMF *Q* (and *Q/Qexp*) values in these columns correspond to the *Q* (and *Q/Qexp*) values associated with the solution space of the respective time-resolved windows only. For details, refer to Sect. 2.2 Eqs. 16–17.

### 3.4.2 Comparison of time series patterns of *Q/Qexp* in different time-of-day periods

We can further explore the time periods and *m/z*s that show improvement in fits in the time-of-day PMF approach. In Fig. 9a–b, we plot the percent change of 15 min averaged *Q/Qexp* values from the seasonal PMF approach to the time-of-day PMF approach in the midday and night-time periods. Monsoon results show limited variability, with the standard deviation (SD) of the percent change less than 5% from the mean (excluding the edges). On the other hand, in winter, the SD of the percent change are ≥15% from the mean, and time-of-day PMF approach particularly improves the solution in the middle of the midday
window (11:30–14:00 LT) and the first half of the night-time window (23:30–00:45 LT). These selective improvements suggest that time-of-day PMF likely accounts for period-specific sources better than the seasonal PMF approach.

### 3.4.3 Comparison of *Q/Qexp* by *m/z* in different time-of-day periods

Instead of classifying improvements in *Q/Qexp* by time, we can classify the improvements by *m/z*s. In Fig. 10a–b, we plot the percent change of *Q/Qexp* at different *m/z*s between the seasonal PMF approach and the time-of-day PMF approach. Our
results show that the percent changes are either negative or small positive at important *m/z* tracers in all periods. In addition, the changes are largely negative at *m/z*s higher than *m/z* 80, suggesting that time-of-day PMF approach particularly improves the fits at *m/z*s higher than *m/z* 80. In particular, winter midday is accompanied by decreases at important *m/z*s such as 29, 41, 43, 44, 55, 57, and 60, as well as *m/z*s higher than *m/z* 80.

       We also observe that the fit quality reduced at some *m/z*s; however, most of these *m/z*s are not tracers of specific PMF
factor types (Zhang et al., 2011). Future work could investigate the deployment of the binPMF approach, selectively fitting important *m/z*s only to identify PMF factors (Zhang et al., 2019). Overall, the time-of-day PMF approach improves PMF fit dissimilarly at different *m/z*s compared to the seasonal PMF approach.

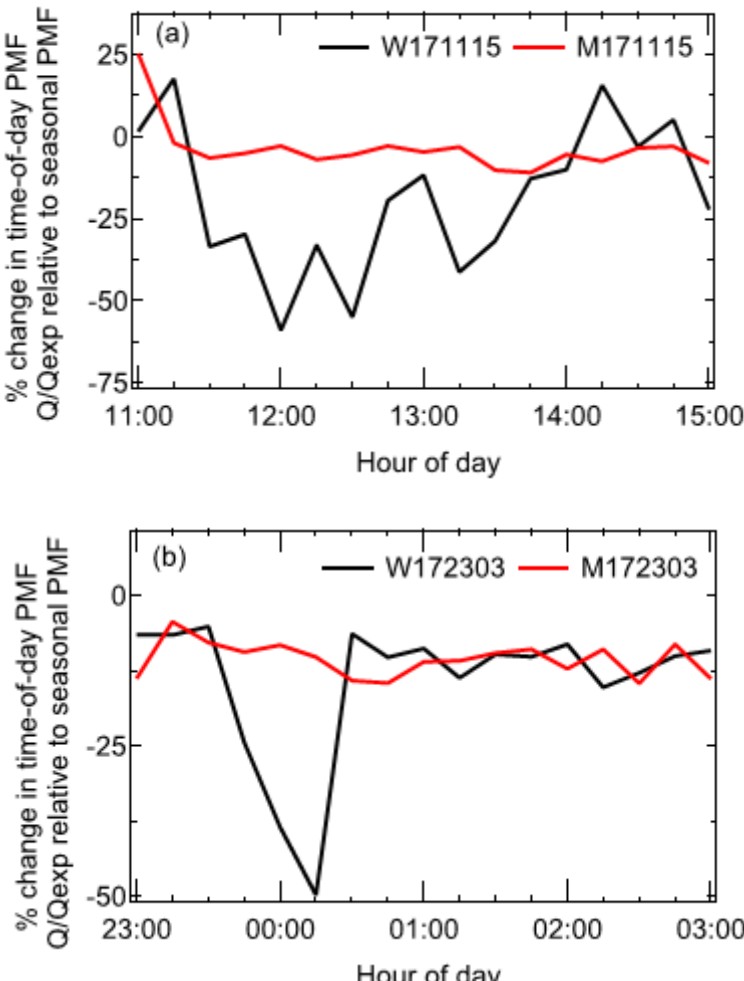

**Figure 9 Percent change of 15 min averaged seasonally representative *Q/Qexp* values between the seasonal PMF approach and the time-of-day PMF approach in (a) midday and (b) night-time periods. Time-of-day PMF selectively improves *Q/Qexp* in specific periods compared to the seasonal PMF approach.**

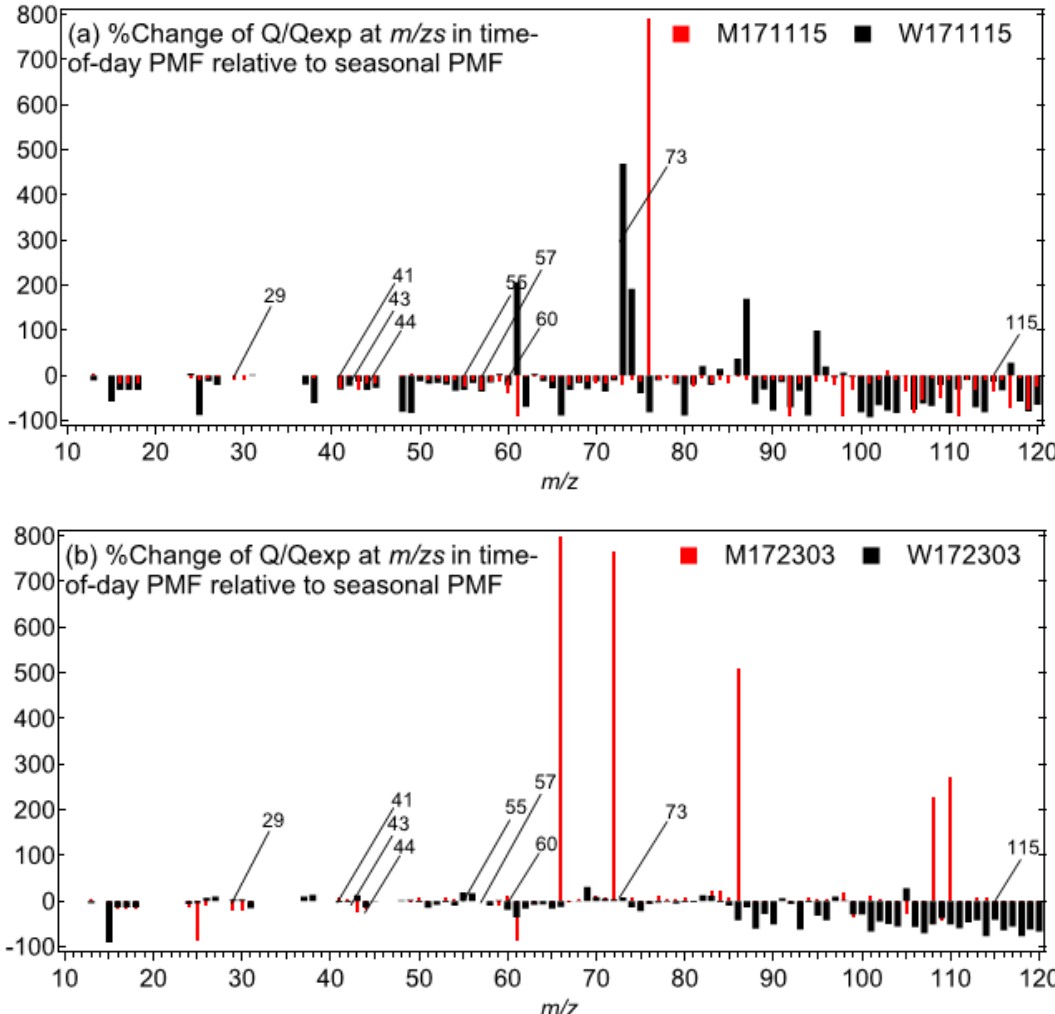

**Figure 10 shows the percent change of *Q/Qexp* at different *m/z*s between the seasonal PMF approach and the time-of-day PMF approach in (a) midday and (b) night-time periods. Key *m/z*s show a lower *Q/Qexp* in the time-of-day PMF approach compared to the seasonal PMF approach.**

**Section 4 Conclusions**

This study introduces a new approach to conducting source apportionment analysis—conducting positive matrix factorization

on long-term datasets with each day separated into six 4 hour periods with limited variability in emissions and meteorology.

The statistical viability of a new source apportionment approach is demonstrated, and the approach is called time-of-day PMF.

We apply the time-of-day PMF approach on two seasons of highly time-resolved speciated non-refractory submicron aerosol

(NR-PM$_1$) organics (Org). This dataset was collected as a part of the Delhi Aerosol Supersite (DAS) study. This study improves


upon the seasonal source apportionment previously employed in Delhi. We use the EPA PMF tool to apply constraints, extract
a larger number of factors, and quantify errors in PMF solutions.

Time-of-day PMF analysis resolves a greater diversity of factors compared to the traditional seasonal PMF approach.
In winter, time-of-day PMF separates a mixed SFC-OA factor and a BBOA factor at midday but separates clean HOA and
BBOA factors at night. Resolving by time-of-day allows the identification of different types of BBOA; the midday BBOA is
associated with chloride and night-time BBOA is associated with black carbon. In monsoon, a mixed COA-HOA factor is
obtained at midday, but separate clean HOA and COA factors are obtained at night. Even the mixed COA-HOA factor shows
clear markers associated with influence of heated cooking oils, especially seen in Asian cooking. Such markers are not seen in
seasonal PMF. PMF analysis also separates two OOA factors in each period, one more local and the other more regional in
nature. The two OOA factors show signs of mixing and are therefore not discussed in detail.

In monsoon, the seasonal PMF approach underestimates POA TS at all times of the day relative to the time-of-day
PMF approach. In winter, the seasonal PMF approach underestimates POA TS at midday, but overestimates POA TS at night.
Several differences also occur at key $m/z$s in POA MS extracted from the two approaches. Time-of-day PMF midday POA
factors are more oxidized than the seasonal PMF POA factors, in line with the high photochemical processing at midday.
Differences in night-time POA MS profiles are small.

OOA TS show strong similarity between the two approaches. However, OOA MS show lower oxidation state at
monsoon midday and winter night-time, and higher oxidation state at monsoon night-time and winter midday in time-of-day
PMF analysis compared to the seasonal PMF analysis. Presence of semi-volatile oxidized organics at monsoon midday and
winter night-time could be attributed to semi-volatile biogenic emissions in monsoon and slow oxidation processes in winter.
$Q/Qexp$ values of the PMF solutions are a measure of quality of fit and show a decrease of 5–30% going from seasonal PMF
analysis to time-of-day PMF analysis. These improvements in $Q/Qexp$ can be mapped out to specific time points and $m/z$s. In
winter, improvements in $Q/Qexp$ are particularly larger in specific time periods in the 4 h time windows. In monsoon, the
improvements are, for the most part, independent of time. in winter, improvements in $Q/Qexp$ are associated with
improvements at key $m/z$s. Improvements in $Q/Qexp$ for all periods are partially driven by improvements in fits at $m/z$s higher
than $m/z$ 80.

Application of PMF on field monitoring datasets is a powerful approach to separate the effects of contributing sources.
Typically, such analysis is conducted on datasets lasting from a few weeks to a few months. However, in the last decade,
several long-term aerosol mass spectrometry deployments have occurred, and one such deployment is the Delhi Aerosol
Supersite study. Long-term measurements are also conducted for regulatory-level air pollution monitoring. In the coming
years, source apportionment strategies could become mainstream policy tools, and organic mass spectrometry instrumentation
may obtain regulatory-grade status. Given this context, the time-of-day PMF approach combines the benefits of large datasets
collected using long-term monitoring with the enhancement of time-resolving capability of source apportionment approaches
such as PMF at a lower computational intensity compared to the traditional approaches. Results in this paper demonstrate that





time-of-day PMF approach gives a greater number of factors as well as more representative PMF factors compared to the traditional seasonal PMF approach.

**Appendix A: Abbreviations**

| | |
|---|---|
| ACSM | Aerosol Chemical Speciation Monitor |
| BBOA | Biomass-burning organic aerosol |
| BC | Black carbon |
| $BC_{BB}$ | wood burning component of BC |
| BS | Bootstrapping |
| BS-DISP | Bootstrapping enhanced with displacement |
| CO | Carbon monoxide |
| COA | Cooking organic aerosol |
| DAS | Delhi Aerosol Supersite |
| DISP | dQ-controlled displacement of factor elements |
| HOA | Hydrocarbon-like organic aerosol |
| IIT | Indian Institute of Technology |
| LT | Local Time |
| ME-2 | Multilinear Engine |
| MS | Mass spectral profiles |
| NCR | National Capital Region |
| $NR-PM_1$ | Non-refractory submicron particulate matter |
| $NR-PM_{2.5}$ | Non-refractory PM smaller than 2.5 µm in diameter |
| OOA | Oxygenated organic aerosol |
| Org | Organic |
| PBLH | Planetary boundary layer height |
| PET | PMF evaluation tool |
| PM | Particulate matter |
| $PM_1$ | Submicron particulate matter |
| $PM_{2.5}$ | Particulate matter smaller than 2.5 µm in diameter |
| PMF | Positive matrix factorization |
| POA | Primary organic aerosol |
| SD | Standard deviation |



| SFC-OA | Solid fuel combustion organic aerosol |
|---|---|
| SOA | Secondary organic aerosol |
| SoFi | Source Finder |
| SVOOA | Semi-volatile oxygenated organic aerosol |
| SWR | Shortwave radiative flux |
| T | Temperature |
| TS | Time Series |
| UVPM | Ultraviolet-absorbing particulate matter |
| VBS | Volatility basis set |
| VC | Ventilation coefficient |

**Data availability**

Underlying research data are available by request to Lea Hildebrandt Ruiz: lhr@che.utexas.edu and will be uploaded to the Texas Data Repository upon acceptance of this manuscript.

**Author contributions**

LHR, JSA, GH, and SB designed the study. SB and ZA carried out the data collection. SB carried out the data processing and analysis. SB, JSA, and LHR assisted with the interpretation of results. All co-authors contributed to writing and reviewing the paper.

**Competing interests**

The authors declare that they have no conflict of interest.

**Acknowledgements**

We are thankful to the Indian Institute of Technology (IIT) Delhi for institutional support. We are grateful to all student and staff members of the Aerosol Research Characterization Laboratory (especially Prashant Soni, Nisar Ali Baig, and Mohammad Yawar) and the Environmental Engineering Laboratory (especially Sanjay Gupta) at IIT Delhi for their constant support. We are thankful to Philip Croteau (Aerodyne Research) for always providing timely technical support for the ACSM, and Dr. Penttti Paatero, Dr. Phil Hopke (University of Rochester), and Dr. Dave Sullivan (UT Austin) for insightful conversations about PMF. I would also like to thank Dr. Nancy Sanchez (now, Chevron; then. at Rice U.) for discussions at the UT Austin





Texas Air Quality Symposium that inspired this work. Lastly, I thank Shahzad Gani (University of Helsinki) for leading the instrument setup for the Delhi Aerosol Supersite study.

**Financial support**

This manuscript is based on work supported in part by the Welch Foundation under Grants F-1925-20170325 and F-1925-20200401 and the National Science Foundation under Grant 1653625. Joshua S. Apte was supported by the ClimateWorks Foundation. We thank the funding agencies for their support.

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
