# Peer review of "Source apportionment resolved by time-of-day for improved deconvolution of primary source contributions to air pollution"

_Atmospheric Measurement Techniques, 2022_

## Author Comment (AC1)

We thank the reviewers for their comments. All comments are addressed below. Reviewers' comments are included in italics, our responses are included in blue, and updated manuscript text is included in red.

**Reviewer 1**

*The manuscript by Sahil Bhandari et al. presented a new method to conduct source apportionment, which can utilize large datasets collected using long-term monitoring compared to traditional positive matrix factorization approaches that do not resolve the diurnal pattern of factor profiles. In addition, the results showed that the new method resolved a greater diversity of factors compared to the traditional seasonal PMF approach in winter and monsoon seasons. In general, this manuscript is well written, but the following aspects should be fully addressed before it can be considered for publication.*

*1. The authors split the data into six 4-hour time windows, and found the differences of MS and TS of OA factors between new method and traditional positive matrix factorization approaches. My major concern is that are these differences (or the characteristics of MS/TS) affected by time division? For example, what are the differences between the results in 11:00-13:00 LT, 13:00-15:00 LT and 11:00–15:00 LT? The authors need to address such uncertainties in the revised manuscript.*

Response: There could be differences in MS/TS within subgroups of data in the four-hour time windows. To address this issue, we conducted detailed uncertainty analysis to ensure that the identified MS/TS are representative of the four-hour time windows (Sect. 2.5). The different uncertainty quantification using the approaches of bootstrapping (BS), displacement (DISP), and bootstrapping enhanced with displacement (BS-DISP) test data subgroups in 100s-10000s of PMF-like model runs to account for the issues of random error and rotational ambiguity within the four-hour time windows (Paatero et al., 2014). Detailed summary statistics from running these uncertainty analyses are presented as mappings onto the PMF solution for the entire time domain (Tables S8-S10). For BS, we set the number of bootstrap resamples at 100 and the default Pearson correlation coefficient at 0.6 for factor assignment. Thus, the BS analysis tests the random error and to a limited extent rotational ambiguity in each four-hour window using 100 additional PMF-like model runs. We show for the chosen combination of factors that BS-mapping, a metric of how factors for subgroups of data in each four-hour window mapped to the factors of the entire time window, is greater than 90% (Table S8). For DISP and BS-DISP, the exact number of PMF-like models conducted with each run depend on the number of species and the number of PMF factors (Paatero et al., 2014). DISP estimates the rotational ambiguity of solutions by displacing MS of each m/z in a factor slightly and attempting to find a solution with changed MS contributions at other m/zs. DISP runs 100s (~150-320) of PMF-like models within each DISP run. BS-DISP

simultaneously tests for random error and rotational ambiguity by first conducting a BS resample and then exploring the rotationally accessible space around each BS resample (Sect. 2.5). Thus, 10000s of (~15000-32000) PMF-like models are run within each BS-DISP run. Instead of mappings, DISP and BS-DISP track factor swaps; a small number of factor swaps suggests low rotational ambiguity and robustness of the PMF solution. We obtained zero swaps in DISP in all cases, and a low count of factor swaps in BS-DISP (Table S9). Thus, by conducting such detailed uncertainty analysis, we have addressed the uncertainties of choosing a given four-hour time window and found PMF factors representative of that four-hour time window, including for subgroups of data. While it is possible that additional information could be obtained using finer time windows, the 4-hour windows seem to have captured diurnal variations of the mass spectra well (e.g., cooking influence midday in contrast to nighttime) while keeping the computational burden under control. Future work could investigate the optimal length of the time window to sufficiently represent the finer time variations (less than 4 hours) in mass spectral profiles while managing computational burden.

To address the reviewer's comments, we have added the following text in Sect. 2.5 Uncertainty estimation:

"This detailed uncertainty analysis ensures that the identified MS/TS are representative of the four-hour time windows by fitting 100s-10000s of PMF-like model runs to data subgroups within the four-hour time windows (Paatero et al., 2014). Detailed summary statistics from running these uncertainty analyses are presented as mappings onto the PMF solution for the entire time domain (Tables S8-S10).The algorithms and computational workload of these techniques are described in detail elsewhere (Paatero et al., 2014)."

*2. More information needs to be listed to support source apportionment results in HOA and COA, as the current mass spectra appear to be confusing. What about their correlations with tracer species? In fact, the authors showed the correlations in Fig. S22-23 and S7-S8, but more discussion should be included in the main text. In addition, how about the results of 4/5/6-factor solutions?*

Response: We address the second half of the reviewer comment first. The detailed decision-making process regarding the number of factors is described in Sect. S1 (Tables S3–S10, Figs. S2–S13, S16–S30). The process is mostly described in the supplement because most of the steps have been documented in established literature (Ulbrich et al., 2009). We discuss the PMF uncertainty analysis in the main manuscript as it has been used relatively sparsely. The EPA PMF tool provides detailed uncertainty analyses tools to validate how representative the chosen PMF solutions are for the respective time windows. Here, we use the uncertainty analysis to select PMF solutions;

we only finalize solutions that pass the EPA PMF tests of random error and rotational ambiguity (Sect. 2.5). All other solutions are rejected. The application of these detailed uncertainty analyses to select a PMF solution for each time window, including the consideration of 4–6-factor solutions, is documented in Table S6, with supporting information in Tables S5, S7–S10.

We separate HOA and COA-related PMF factors in multiple periods. In monsoon seasonal PMF analysis, we separate an HOA factor with an MS strongly correlated with the reference COA factor MS. However, for separation of cooking organic aerosol in this study, we used the Robinson et al (2018) ratio of contributions at m/z 55:57 of 1.6 as a preliminary test for relative positioning of the HOA and COA profiles (COA factors with the ratio close to or greater than 1.6 and HOA profiles with the ratio substantially lower than 1.6). The monsoon seasonal POA factor MS had a m/z 55 to m/z 57 ratio of 1.2 (Fig. S5). Therefore, the seasonal monsoon POA factor is presented as an HOA factor. In M172303, we observe clear separation in the MS of HOA and COA factors, with strong correlations (R>0.9) with respective reference MS profiles, and ratio of contributions at m/z 55:57 of 1.7 for the COA factor and 1.1 for the HOA factor (Fig. S20).  In M171115, the COA-HOA factor mass spectra might appear confusing since the factor MS is strongly correlated to both HOA and COA reference MS profiles. Indeed, the ratio of contributions at m/z 55:57 of 1.4 for the COA-HOA factor reflects factor mixing. In W171115, the SFC-OA profile obtained correlates strongly with a solid-fuel combustion profile obtained using measurements in Delhi elsewhere (Tobler et al., 2020; correlation at all m/zs but m/z 44, Pearson R>0.95, Fig. S18). Also, the ratio of contributions at m/z 55:57 of 1.1 for the SFC-OA factor reflects the limited influence of cooking on that factor.

We present tracer species and their time series correlations with PMF factors in Figs. S6–S8 (seasonal PMF analysis) and Figs. S21–S24 (time-of-day PMF analysis). We use two tracers for HOA-influence: CO and the fossil-fuel component of black carbon, BCFF, estimated using the model of Sandradewi et al. (2008). For the time-series of BBOA factors, we use three tracers: (i) chloride (under the influence of agricultural and other open waste burning-related contributions (Li et al., 2014a, b; Kumar et al., 2015; Fourtziou et al., 2017), (ii) ΔC, defined as the difference between UVPM (370 nm) and BC detected by the aethalometer (Wang et al., 2011; Olson et al., 2015; Tian et al., 2019), and (iii) the biomass-burning component of black carbon, BCBB, estimated using the model of Sandradewi et al. (2008). COA-related factors often exhibit weak correlations with external tracers (Huang et al., 2010, Sun et al., 2011, Liu et al., 2012, Sun et al., 2013, Hu et al., 2016, Stavroulas et al., 2019). However, correlations with chloride in COA-like factors is suggestive of the influence of landfill emissions, trash burning, and solid-fuel sources (Dall'Osto et al., 2015, Lin et al., 2017). In the seasonal monsoon PMF run (M17), we observe only one primary factor, an HOA factor with strong correlations with tracers CO (Spearman R: 0.73) and BCFF

(Spearman R: 0.91) (Fig. S8). The winter midday SFC-OA profile correlates strongly with chloride (Spearman R: 0.71), nitrate (Spearman R: 0.75), BCFF (Spearman R: 0.79), and ΔC (Spearman R: 0.60), pointing to the mixing of HOA, BBOA, and possibly COA influence in the factor (Fig. S21). At winter nighttime, we separate an HOA MS profile that correlates strongly with BCFF (Spearman R: 0.84) and CO (Spearman R: 0.83). We obtain one BBOA factor each at winter midday and winter nighttime. Among the two BBOA obtained, winter midday BBOA correlates strongly with chloride (Spearman R: 0.66) and CO (Spearman R: 0.67), suggesting an industrial source (Fig. S21, Sect. 3.1). At nighttime however, winter BBOA correlates strongest with the wood burning component of BC (BCBB, Spearman R: 0.92) and weakly with chloride (Spearman R: 0.40), suggesting at least two different origins of BBOA (Fig. S22). This is consistent with our previous work, where we have separated BBOA-like factors with different correlations with chloride and BCBB in different seasons (Bhandari et al., 2020; Patel et al., 2021a). In monsoon midday, we observe only one primary factor, a COA-HOA factor, with strong correlations with chloride (Spearman R: 0.75), suggesting the influence of landfill emissions, trash burning, and solid-fuel sources (Fig. S23). Otherwise, COA-HOA has weak correlations with external tracers. In the monsoon nighttime PMF run (M172303), we observe stronger correlations of the HOA factor with CO (Spearman R: 0.79) and BCFF (Spearman R: 0.86) compared to correlations of these tracers with the COA factor (CO: Spearman R: 0.70, BCFF: Spearman R: 0.71) (Fig. S24).

To address the reviewer comments regarding methods to obtain PMF solutions, including checking for 4/5/6-factor solutions, we have updated the text in Sect. 2.4:

"Details of the steps for conducting PMF, R code, and criteria for factor selection are discussed in detail in the Supplement (Sect. S1). Briefly, for selection of PMF solutions, we started by analyzing the different statistics of Q/Qexp (a measure of fit), correlogram of residual TS and correlation with external tracers, time series patterns in residuals, and PMF fits at different m/zs (Table S4). We also considered the correlation of factor mass spectral profiles with reference mass spectra since MS of different factors are characterized by different spectral signature peaks (Zhang et al., 2011). For example, hydrocarbon-like organic aerosol (HOA) is a proxy for fresh traffic and combustion emissions and shows prominent peaks at $m/z$ values 55 and 57 and a higher fractional organic signal at $m/z$ 43 than $m/z$ 44. For separation of cooking organic aerosol (COA) and distinguishing it from HOA in this study, we used the Robinson et al (2018) ratio of contributions at m/z 55:57 of 1.6 as a preliminary test for relative positioning of the HOA and COA profiles (COA factors with the ratio close to or greater than 1.6 and HOA profiles with the ratio substantially lower than 1.6). We also validated obtained PMF factors by correlation of factor time series with external tracers. We use two tracers for

HOA-influence: CO and the fossil-fuel component of black carbon, $BC_{FF}$, estimated using the model of Sandradewi et al. (2008). For the time-series of BBOA factors, we use three tracers: (i) chloride (under the influence of agricultural and other open waste burning-related contributions (Li et al., 2014a, b; Kumar et al., 2015; Fourtziou et al., 2017), (ii) ΔC, defined as the difference between UVPM (370 nm) and BC detected by the aethalometer (Wang et al., 2011; Olson et al., 2015; Tian et al., 2019), and (iii) the biomass-burning component of black carbon, $BC_{BB}$, estimated using the model of Sandradewi et al. (2008). COA-related factors often exhibit weak correlations with external tracers (Huang et al., 2010, Sun et al., 2011, Liu et al., 2012, Sun et al., 2013, Hu et al., 2016, Stavroulas et al., 2019). Additionally, the EPA PMF tool provides detailed uncertainty analyses tools to validate how representative the chosen PMF solutions are for the respective time windows. Here, we use the uncertainty analysis to select PMF solutions; we only finalize solutions that pass the EPA PMF tests of random error and rotational ambiguity, as described below in Sect. 2.5. The application of these detailed uncertainty analyses to select a PMF solutions for each time window, including the consideration of 3–8-factor solutions, is documented in Table S6, with supporting information in Tables S5, S7–S10."

To address the reviewer's comments regarding time series correlations with tracer species, we have updated the text in Sect. 3.1:

"In monsoon, the seasonal PMF HOA MS is also strongly correlated with the reference COA factor MS (Ng et al., 2011a; Pearson R~0.90; Fig. S5). However, the monsoon seasonal POA factor MS had a m/z 55 to m/z 57 ratio of 1.2 (Fig. S5). Therefore, the seasonal monsoon POA factor is presented as an HOA factor. This HOA factor has stronger correlations with tracers CO (Spearman R: 0.73) and BCFF (Spearman R: 0.91) than the OOA factors (Fig. S6)."

To address the reviewer's comments regarding time series correlations with tracer species, we have also updated the text in Sect. 3.2.1:

"The winter midday SFC-OA profile correlates strongly with chloride (Spearman R: 0.71), nitrate (Spearman R: 0.75), $BC_{FF}$ (Spearman R: 0.79), and ΔC (Spearman R: 0.60), pointing to the mixing of HOA, BBOA, and possibly COA influence in the factor (Fig. S21). At winter nighttime, we separate an HOA MS profile that correlates strongly with BCFF (Spearman R: 0.84) and CO (Spearman R: 0.83). We obtain one BBOA factor each at winter midday and winter nighttime. Among the two BBOA obtained, winter midday BBOA correlates strongly with chloride (Spearman R: 0.66) and CO (Spearman R: 0.67), suggesting an industrial source (Fig. S21, Sect. 3.1). At nighttime however, winter BBOA correlates strongest with the wood burning component of BC

(BC$_{BB}$, Spearman R: 0.92) and weakly with chloride (Spearman R: 0.40), suggesting at least two different origins of BBOA (Fig. S22). This is consistent with our previous work, where we have separated BBOA-like factors with different correlations with chloride and BC$_{BB}$ in different seasons (Bhandari et al., 2020; Patel et al., 2021a). In monsoon midday, we observe only one primary factor, a COA-HOA factor, with strong correlations with chloride (Spearman R: 0.75), suggesting the influence of landfill emissions, trash burning, and solid-fuel sources (Fig. S23). Otherwise, COA-HOA has weak correlations with external tracers. In the monsoon nighttime PMF run (M172303), we observe stronger correlations of the HOA factor with CO (Spearman R: 0.79) and BC$_{FF}$ (Spearman R: 0.86) compared to correlations of these tracers with the COA factor (CO: Spearman R: 0.70, BC$_{FF}$: Spearman R: 0.71) (Fig. S24)."

*3. What is the justification for distinguishing between local OOA and regional OOA? Figs. S27–S30 did not support your conclusion in lines 412-414 in my sense.*

Response: In Figs. S27–S30 we present the normalized level diurnal variations of the local OOA and the regional OOA factors. Typically, regional OOA is more oxidized (shows weaker correlations with reference SVOOA MS) and has less diurnal variation, in line with its expected average lower volatility and contributions from long-range transport (Drosatou et al., 2019). To quantify the flatness of the diurnal variations in each time window, the table below presents the lowest and the highest levels (relative to the normalization to 1) and the range (difference of the highest to the lowest levels) of the two factors. In all periods, the local OOA exhibits a similar or larger range than the regional OOA factor. However, we see an overlap of the 95% confidence intervals of the normalized levels (Figs. S27–S30).

| Period | Lowest level | | Highest level | | Range (Highest − Lowest) | |
|---|---|---|---|---|---|---|
| | Local OOA | Regional OOA | Local OOA | Regional OOA | Local OOA | Regional OOA |
| W171115 | 0.86 | 0.82 | 1.21 | 1.08 | 0.35 | 0.26 |
| W172303 | 0.87 | 0.71 | 1.26 | 1.11 | 0.39 | 0.40 |
| M171115 | 0.81 | 0.89 | 1.18 | 1.15 | 0.37 | 0.26 |
| M172303 | 0.79 | 0.97 | 1.12 | 1.02 | 0.33 | 0.05 |

We have updated the text (Sect. 3.2.2):

"Time-of-day PMF and seasonal PMF generate two OOA factors, local OOA and regional OOA, in each run (Figs. S25 and S26). Typically, regional OOA is more oxidized (shows weaker correlations with reference SVOOA MS) and has less diurnal variation, in line with its expected average lower volatility and contributions from longrange transport (Drosatou et al., 2019). The time-of-day PMF OOA factors show MS and TS behavior similar to the seasonal PMF OOA factors, as shown in Sect. 3.3. Mass spectra of both local OOA and regional OOA correlate strongly with the reference OOA factor (Pearson R>~0.80) (Figs. S25 and Figs. 26). Also, we consistently observe that the more oxidized regional OOA factors have flatter diurnal time series patterns (smaller range) than the less oxidized local OOA factors (larger range) (Figs. S27–S30; Table S11). However, we see an overlap of the 95% confidence intervals of the normalized levels (Figs. S27–S30) and an overlap of external tracers suggesting mixing of the two OOA components (see Sect. S4). This is not surprising considering similarity of the MS of the two OOA factors and a continuum of the level of oxidation in the atmosphere. Since we observe factor mixing of the two secondary components, detailed analysis of the factor MS and TS (correlations with external tracers, features of the mass spectra) are only presented in the Supplement (see Sect. S4)."

We have also added the table to the Supplement (Table S11).

*4. What are the correlations of same type of OA factors between the daytime and nighttime? It would be nice to have some comparison of MS of the same type of OA factors between daytime and nighttime. In my sense, the differences in MS between day and night in OA factors are the highlights of this paper. However, the potential differences between day and night and the reasons have not been discussed in depth.*

Response: We agree with the reviewer that the differences in MS between day (midday) and night (nighttime) in OA factors are key findings of this paper. Currently, we discuss these differences in Sect. 3.3, by combining all POA factors into one POA factor and all OOA factors into one OOA factor in each period. We report the mass spectral correlations for the OOA factors in Figs. S25-S26 and MS comparisons in Figs. S31-S32 and S37-S38. Time-of-day PMF analysis is able to capture differences between day and night MS because it conducts PMF analysis for each period independent of the influence of variability in the other periods. This has been discussed in detail in Sect. 3.3, where the time-of-day PMF MS and TS are compared to seasonal PMF MS and TS.

In the revised manuscript, we have created a separate subsection to discuss the daytime and nighttime MS comparisons, and also added a brief description of the reasons for the differences. We have also added and discuss Figs. S39-S40, showing the MS correlations of the midday and nighttime POA factors across the two techniques, along with their correlations to reference MS profiles. We have updated the text (Sect. 3.4):

"However, this contrast is sharper in time-of-day PMF analysis, in line with the ability of the approach to capture variable MS (Figs. S31b and S32b; Fig. S39, Winter Time-of-day POA Spearman R: 0.93, Winter Seasonal POA Spearman R: 0.97; Fig. S40, Monsoon Time-of-day POA Spearman R: 0.81, Monsoon Seasonal POA Spearman R: 1.0). The seasonal PMF midday–night-time comparison also fails to capture the influence of cooking midday based on the low and similar ratio of contributions at m/z 55 to m/z 57 as night-time, esp. in monsoon (~1, Figs. S31a and S32a). This contrast between midday and night-time POA MS is higher in time-of-day PMF in winter (midday ratio: 1.2, night-time ratio: 1.0, Fig. S31b) and in monsoon time-of-day PMF analysis (midday ratio: 1.4, night-time ratio: 1.2, Fig. S32b). While seasonal PMF analysis for monsoon suggests no change in MS between midday and night-time, time-of-day PMF analysis suggests large shifts in contributions at key m/zs such as 41, 43, 44, 55, and 57, in line with the changing importance of cooking from midday to night. These differences demonstrate the ability of time-of-day PMF to capture variable MS corresponding to the source influence of those time-of-day periods (Sect. 3.3.2).

We can also compare OOA MS and TS as well as conduct midday and night-time comparisons for time-of-day PMF and seasonal PMF analysis (Sect. S4). Time-of-day PMF OOA MS and TS are similar to seasonal PMF OOA (Table 3, TS: Pearson R>0.95; Figs. S25-S26 MS: Pearson R≥0.95). However, the mass spectra of the time-of-day PMF OOA have major differences at m/z 44 relative to the seasonal PMF OOA (Figs. S35–S36a–b). Comparisons of midday and night-time time-of-day PMF OOA MS shows interesting patterns not apparent in seasonal PMF analysis (Figs. S37–S38a–b). For example, time-of-day PMF analysis for monsoon 2017 suggests less oxidized OOA at midday than night-time, likely caused by the presence of semi-volatile compounds (Fig. S38b). Similar behaviour has been observed elsewhere as well, and was attributed to biogenic emissions (Canonaco et al., 2015).

Figure S42 shows all PMF factors obtained in this paper on the triangle plot (Ng et al., 2010). We observe that factors obtained in the time-of-day PMF analysis occupy a larger spread compared to those obtained in seasonal PMF analysis. For example, in time-of-day PMF POA factors, we observe a spread of about 5% in contributions at m/z 43. In contrast, the spread of seasonal PMF POA factors is less than 3%. Overall, because time-of-day PMF conducts PMF analyses for each period independent of the influence of the variability in the other periods, it generates more representative MS for each time-of-day period (Sect. 3.3)."

Since we have created a new subsection, we have updated the text elsewhere as well (Sect. 3):

"In Sect. 3.4, we compare the midday and nighttime POA and OOA MS profile results from the seasonal PMF and the time-of-day PMF approach. Our hypothesis is that the time-of-day PMF approach will show larger variability across the two time periods. In

Sect. 3.5, we discuss period-specific *Q* (and *Q/Qexp*) values for the time-of-day PMF approach and the seasonal PMF approach. We also compare the *Q/Qexp* TS patterns and *Q/Qexp* by *m/z* to identify periods and *m/z*s with particularly significant changes in *Q/Qexp*."

*5. Zoom the legend in axis in Fig. S22-23 and S7-S8, so that the readers can see them clearly.*

Response: For accessing the expanded figures corresponding to these supplemental figures, we have provided corresponding Supplementary Files, which have larger axis-legends.

*5. Repeated descriptions: lines 216-217 and lines 159-162.*

Response: We have moved lines 159-162 to Sect. 2.4 and removed the repeated descriptions in lines 216–217 in the updated manuscript.

References:

[revised manuscript text omitted]

---

## Author Comment (AC2)

We thank the reviewers for their comments. All comments are addressed below. Reviewers' comments are included in italics, our responses are included in blue, and updated manuscript text is included in red.

*Reviewer 2*

*General Comments:*

*The manuscript by Bhandari et al. presents an innovative application of the PMF for long term and highly time resolved datasets. The fact that various sources influence a site at specific hours throughout the day, running PMF at different time of the day appears to be a logical approach, as it also allows for more MS variability. The application of this time-of-day PMF on a long term ACSM dataset (Delhi, winter and monsoon 2017) improved the source apportionment of OA, by further separating source-specific POA compared to results obtained with standard seasonal PMF. This paper is clearly written and relatively well structured. Some minor comments need to be addressed before being accepted.*

*Minor Comments:*

*1. "Results from PMF analysis for all times of the day are presented in a companion paper (Bhandari et al., 2022)." I find that at least a brief overview of the different factors observed for all time-of-day results should be described in the supplement. Indeed, the change in POA factors from non sequential time-of-day, here 11am-15pm and 23pm-3am, is possible assuming dilution, atmospheric processing, or drastic change in air masses influencing the site. While, for time-of-day following one another (e.g. 11am-3pm and 3pm-7pm), I wonder if POA factors and their concentrations show a decrease before disappearing at a later timing (e.g after 7pm). Reconstructing the diurnal profiles of all POA and SOA of all time-of-day results compared to seasonal one could support the advantage of the new approach.*

Response: We agree with the reviewer that reconstructing the diurnal profiles of POA and SOA factors could show the continuities in time-of-day PMF factor concentrations and thus, the advantage of the new approach. In the companion publication, we generate diurnal profiles for reconstructed POA and OOA in the winter and monsoon seasons (Figs. 2–3, Figs. 4, 7 in Bhandari et al, 2022). We adapt the figures in the companion paper to present diurnal patterns for POA and OOA in the two seasons in the Supplement of this manuscript. In addition, we recently submitted the final response to the companion paper, and as such we expect the two papers to be published at a similar time. In line with the reviewer's suggestion, we have updated the text (Sect. 2.4):

"Results from PMF analysis for all times of the day are presented in a companion paper (Bhandari et al., 2022), and a brief summary of those results is also provided in the Supplement (Sect. S5)."

The text above was moved to Sect. 2.4 to address another reviewer's comment. We have also updated the Supplement (Sect. S5):

"Results from PMF analysis for all times of the day are presented in a companion paper (Bhandari et al., 2022). Here, we share a brief summary of those results, focusing on diurnal patterns of POA and OOA in seasonal PMF and time-of-day PMF. Figure S41a-b show the diurnal time series patterns of POA (HOA+BBOA+COA) and OOA (Local OOA + Regional OOA) factors for winter and monsoon of 2017. Clearly, POA concentrations exhibit larger variability than OOA concentrations in both seasons. Our results show that the time series (TS) concentrations of time-of-day PMF factors are broadly consistent with seasonal PMF factors. In winter, we separated BBOA or BBOA-like factors in all periods but did not separate cooking organic aerosol (Table S3 in Bhandari et al., 2022). We also separated HOA or HOA-like factors in all time-of-day periods in winter. In monsoon 2017, we separated HOA or HOA-like factors, and COA or COA-like factors in all time-of-day periods but did not separate biomass burning organic aerosol above detection limits (Tables 2, S3 in Bhandari et al., 2022). The behaviour of POA and OOA TS obtained by combining all time-of-day PMF results suggests strong similarities to seasonal PMF POA and OOA TS, respectively (W17 POA: slope ~ 0.83, intercept ~ 1.6, $R$~0.97; W17 OOA: slope ~ 1.26, intercept ~ −7.0, $R$ ~ 0.88; M17 POA: slope ~ 1.15, intercept ~ 1.5, $R$~0.97; M17 OOA: slope ~ 0.91, intercept ~ −0.5, $R$ ~ 0.98). In winter, we observe largest differences in POA TS diurnal concentrations midday where primary concentrations are higher in time-of-day PMF by ≥40%. Because of the low total OA concentrations in these periods, they likely have limited importance in seasonal PMF analysis with respect to determining the overall seasonal mass spectra and time series patterns, and thus conducting time-of-day PMF analysis results in factors exhibiting substantial deviations from seasonal analysis. In monsoon, seasonal PMF analysis underestimates POA concentrations throughout the day. Finally, we also observe that winter time-of-day PMF OOA time series patterns exhibit significantly lower diurnal variability than time-of-day PMF POA but stronger diurnal variability than seasonal PMF OOA. For the time-of-day PMF approach, winter peak OOA diurnal concentrations in the morning (0900–1000 hours) are ~2.7 times the diurnal minimum (which occurs in the evening, 1800–1900 hours); substantially greater than the ~2.2 observed for seasonal PMF winter OOA concentrations. This difference is driven by lower OOA concentrations midday (1100–1900 hours) and higher OOA concentrations at other hours. In monsoon, OOA concentrations show similar diurnal patterns between time-of-day PMF and seasonal PMF and OOA concentrations are almost always lower in time-of-day PMF. Clearly, time-of-day PMF captures different aspects of diurnal variability better than seasonal PMF, which is a major advantage of this new approach."

*2. Table S2: how do you explain the differences in term of mass concentrations for OOA during W172303 even though similar factors are identified with both PMF type?*

Response: We agree with the reviewer that similar factors are identified during W172303 using seasonal PMF analysis and time-of-day PMF analysis (Table 2). In addition, MS and TS

patterns of the respective factors are also very similar (Sect. 3.3). While a comparison of MS shows that the two techniques generate similar MS, we also note that they apportion substantially different mass contributions of an important m/z, m/z 44, in OOA MS (about 4% larger organic mass at m/z 44 in seasonal PMF OOA, Fig. S35). We also see differences in the POA MS (Fig. 7), but the difference in MS is much smaller (about 1% smaller organic mass at m/z 44 in seasonal PMF POA). This change between the apportionment of the two techniques means that for similar total apportioned concentrations at m/z 44 (which is fixed and based on the underlying data common to the two approaches), the time series concentrations for the time-of-day PMF OOA would end up larger, which is the case (Table 2). Nevertheless, we also show that the time series concentrations for W172303 OOA based on the two techniques are strongly correlated (Table 3; Pearson R of 0.98).

*3. Page 7 line 215: You mentioned that focusing on the 11am-15pm time of day "we expect to see more oxidized aerosols". Two SOA were identified regardless of the type of PMF applied. Is the ACSM mass resolution limiting further separation or could it be that some of the seasonal SOA are identified as oxidized POA in the time-of-day PMF (e,g oxidized BBOA)?*

Response: Yes, conducting PMF on the midday period leads to the same two SOA factors identified regardless of the type of PMF applied. Additionally, changes in MS between the seasonal PMF and time-of-day PMF analysis result in mass moving from seasonal OOA to POA in the midday PMF windows (Table 2). Also, in line with the rapid photochemical processing midday, we observe more oxidized POA in time-of-day PMF than seasonal PMF (Figs. 7, 8). However, in both seasons, we observe strong correlations of these POA factor TS with primary tracers (Figs. S21, S23) and POA MS with reference POA MS profiles (Figs. S16, S19). We also conducted detailed uncertainty analysis to ascertain the validity of our PMF solutions (Sect. 2.5). Thus, the seasonal SOA likely have minimal influence on the identified oxidized POA factors, given the evidence of POA MS and TS signatures of these factors. Instead, we believe that time-of-day PMF analysis provides a more realistic set of MS and TS patterns than the seasonal PMF analysis, given that by design, time-of-day PMF analyses conducts PMF analysis for each period independent of the influence of the variability in the other periods. The deployment of higher mass-resolution instrumentation such as EESI-TOF allows separation of specific SOA factors, even in seasonal PMF analysis (Stefenelli et al., 2019). This observation suggests that the ACSM mass resolution might limit further separation. Also, in this work, no constraints were imposed on the presence of more detailed secondary organic aerosol factors, such as cooking SOA or traffic SOA. Future work could explore constraining the presence of these factors in ACSM-based PMF analysis.

*4. Page 7 line 222: "Future work should investigate the optimal length of the time window to sufficiently represent the diurnal variations in mass spectral profiles while managing computational burden". I also think that more explanations should be provided regarding your choice of using a 4 hours window and to focus on 11-15 and 23-03.*

Response: We address the second part of the reviewer's comment first. As shown in Table 1 and Fig. S1, and described in Sect. 2, the two periods allow differentiation between two

extremes in terms of reaction chemistry and meteorology. Midday periods typically have rapid photochemical processing and higher temperatures compared to nighttime periods. Additionally, they also differ in the influencing primary sources in those time periods; midday is expected to have a stronger influence of cooking, and nighttime of biomass burning. Additionally, as shown in the companion paper, these periods represent the two extremes in total NRPM1 concentrations (Tables 1–2, Bhandari et al., 2022).

The choice of the four-hour window was based on a preliminary PMF analysis conducted on monsoon that allowed us to identify the influence of cooking organic aerosol in the midday PMF run, based on the Robinson et al (2018) ratio of contributions at m/z 55:57 of 1.6 as a preliminary test for relative positioning of the HOA and COA profiles (COA factors with the ratio closer or greater than 1.6 (Bhandari et al., AAAR, 2019). We started from 12hour time windows and kept decreasing the window size until the ratio was significantly greater than expected for an HOA factor. We settled for a four-hour time window to limit computational burden and the number of PMF runs needed to cover all times of the day.

We have updated the manuscript by moving lines from Sect. 2.2 to Sect. 2.4 and have updated the text of these lines:

"Here, we used two alternative approaches for conducting PMF. In one approach, we apply PMF by splitting the data into six 4-hour time windows each day to illustrate the use of our time-of-day PMF method. The choice of the four-hour window was based on a preliminary PMF analysis conducted on monsoon that allowed us to identify the influence of cooking organic aerosol, based on the ratio of contributions at m/z 55:57 (Robinson et al, 2018). We started from 12 hour time windows and kept decreasing the window size until the ratio was substantially greater than 1.6, suggesting the presence of a COA factor in at least one such time window (in this case, it was M172303, Table 2). We also conduct seasonal PMF runs for winter and monsoon 2017 and time-of-day PMF runs for two periods (1100–1500 LT-local time and 1900–2300 LT) in the two seasons. Thus, we conduct four time-of-day PMF runs in total. The two time-of-day periods are selected to differentiate between influence of primary sources, changing MS due to reaction chemistry, and effect of meteorology (Table 1, Fig. S1). As shown in the companion paper, these periods represent the two extremes in total NRPM1 concentrations (Tables 1–2, Bhandari et al., 2022). Results from PMF analysis for all times of the day are presented in a companion paper (Bhandari et al., 2022), and a brief summary of those results is also provided in the Supplement (Sect. S5). In monsoon and winter, traffic is expected to be a dominant source at night due to low cooking-related emission and overlap with high night-time traffic on major traffic corridors (Mishra et al., 2019). At midday in monsoon, high temperatures and solar flux imply high photochemical processing of aerosols; therefore, we expect to see more oxidized aerosols (Table 1, Fig. S1). At winter night-time, biomass burning for heating is an expected source. We refer to the seasonal organic MS-based PMF analysis results as "seasonal PMF" and time-of-day organic MS-based PMF analysis results as "time-of-day PMF" results in the paper. To refer to PMF runs corresponding to specific time windows, we use the nomenclature "Season" + "Period" style in the format "STTTT" (Table 1). For example, W1115 corresponds to the 1100–1500 LT of Winter 2017.

Using data presented in this paper, we also compare the Q (and Q/Qexp) values from the seasonal PMF runs corresponding to the periods of the time-of-day windows (Sect. 3.5). While this work addresses the diurnal variations in MS patterns, future work could investigate the optimal length of the time window to sufficiently represent the finer time variations (less than 4 hours) in mass spectral profiles while managing computational burden."

*6. Adding the F44 vs F43 diagram could help segregating the different type of OA.*

Response: We show the triangle plot (Ng et al., 2010) for the different PMF factors presented in this paper in Figure S42. Broadly, we see that almost all factors lie within the plot. Also, broadly, the primary factors occupy the lower portion of the triangle plot (0.05<f43<0.12, f44<0.07), and secondary factors occupy a narrow region of the plot in the top left (f44>0.1, 0.07<f43<0.09). We also observe that factors obtained in the time-of-day PMF analysis occupy a larger spread compared to those obtained in seasonal PMF analysis. For example, in time-of-day PMF POA factors, we observe a spread of about 5% in contributions at m/z 43. In contrast, the spread of seasonal PMF POA factors is less than 3%. Future work could utilize cluster analysis and other dimensionality reduction techniques on the distribution of f44 versus f43 in ambient data to identify sources (Isokääntä et al., 2020; Koss et al., 2020; Liang et al., 2021; Ma et al., 2022).

[Figure]

*7. Figure 3 and later: keep consistent writing of the unit "µg m-3" in text/captions/figures (main text and SI).*

Response: We have updated the text to reflect this change.

*8. Lines 389-390: change "at" to "in the afternoon".*

Response: We have updated the following text (Sect. 3.2.1):

"These high contributions are likely a result of the highly oxidizing environment in the afternoon."

*9. I think that the different MS identified for the time-of-day PMF would add more value to the discussion and would be more useful in section 3.2.1 and 3.2.2 instead of having them in the SI.*

Response: We agree with the reviewer that the obtained MS identified add value to the discussion and would be useful in the main manuscript. However, MS obtained for all PMF factors using the time-of-day PMF approach for all periods in winter and monsoon have been documented in the companion publication (Bhandari et al., 2022). As such, the main purpose of this manuscript is to document the development of a new approach to conducting PMF, the time-of-day PMF approach, and to validate the approach relative to the traditional seasonal PMF approach. We believe that the mass spectral comparisons shown in Figs. 7–8 , S31–S32, and S35-S38 document the most important MS identified in this work. We believe that the brief summary of results from the companion paper provided in the Supplement (Sect. S5), and the multiple references to the companion paper will encourage the readers to read the two manuscripts together to fully understand the scope of this work.

References

1. Bhandari, S., Arub, Z., Habib, G., Apte, J. S., and Ruiz, L. H.: Contributions of primary sources to submicron organic aerosols in Delhi, India, https://acp.copernicus.org/preprints/acp-2022-179/, submitted to Atmospheric Chemistry and Physics, 2022..

2.Bhandari, S., Patel, K., Gani, S., Habib, G., Apte, J., Ruiz, L. H.: Application of Advanced Factorization Techniques for Deconvolution of Cooking and Biomass Burning Source Contributions in a Polluted Megacity: American Association for Aerosol Research, Portland, OR, USA, https://aaarabstracts.com/2019/viewabstract.php?pid=523, 2019.

3.Isokääntä, S.; Kari, E.; Buchholz, A.; Hao, L.; Schobesberger, S.; Virtanen, A.; Mikkonen, S. Comparison of Dimension Reduction Techniques in the Analysis of Mass Spectrometry Data. *Atmospheric Measurement Techniques* **2020**, *13* (6), 2995–3022. https://doi.org/10.5194/amt-13-2995-2020.

4. Koss, A. R.; Canagaratna, M. R.; Zaytsev, A.; Krechmer, J. E.; Breitenlechner, M.; Nihill, K. J.; Lim, C. Y.; Rowe, J. C.; Roscioli, J. R.; Keutsch, F. N.; Kroll, J. H. Dimensionality-Reduction Techniques for Complex Mass Spectrometric Datasets: Application to Laboratory Atmospheric

Organic Oxidation Experiments. *Atmospheric Chemistry and Physics* **2020**, *20* (2), 1021–1041. https://doi.org/10.5194/acp-20-1021-2020.

5. Liang, Y.; Jen, C. N.; Weber, R. J.; Misztal, P. K.; Goldstein, A. H. Chemical Composition of PM2.5 in October 2017 Northern California Wildfire Plumes. *Atmospheric Chemistry and Physics* **2021**, *21* (7), 5719–5737. https://doi.org/10.5194/acp-21-5719-2021.

6. Ma, J.; Ungeheuer, F.; Zheng, F.; Du, W.; Wang, Y.; Cai, J.; Zhou, Y.; Yan, C.; Liu, Y.; Kulmala, M.; Daellenbach, K. R.; Vogel, A. L. Nontarget Screening Exhibits a Seasonal Cycle of PM2.5 Organic Aerosol Composition in Beijing. *Environ. Sci. Technol.* **2022**, *56* (11), 7017–7028. https://doi.org/10.1021/acs.est.1c06905.

7. Ng, N. L.; Canagaratna, M. R.; Zhang, Q.; Jimenez, J. L.; Tian, J.; Ulbrich, I. M.; Kroll, J. H.; Docherty, K. S.; Chhabra, P. S.; Bahreini, R.; Murphy, S. M.; Seinfeld, J. H.; Hildebrandt, L.; Donahue, N. M.; DeCarlo, P. F.; Lanz, V. A.; Prévôt, A. S. H.; Dinar, E.; Rudich, Y.; Worsnop, D. R. Organic Aerosol Components Observed in Northern Hemispheric Datasets from Aerosol Mass Spectrometry. *Atmospheric Chemistry and Physics* **2010**, *10* (10), 4625–4641. https://doi.org/10.5194/acp-10-4625-2010.

8. Robinson, E. S.; Gu, P.; Ye, Q.; Li, H. Z.; Shah, R. U.; Apte, J. S.; Robinson, A. L.; Presto, A. A. Restaurant Impacts on Outdoor Air Quality: Elevated Organic Aerosol Mass from Restaurant Cooking with Neighborhood-Scale Plume Extents. *Environ. Sci. Technol.* **2018**, *52* (16), 9285–9294. https://doi.org/10.1021/acs.est.8b02654.

9. Stefenelli, G.; Pospisilova, V.; Lopez-Hilfiker, F. D.; Daellenbach, K. R.; Hüglin, C.; Tong, Y.; Baltensperger, U.; Prévôt, A. S. H.; Slowik, J. G. Organic Aerosol Source Apportionment in Zurich Using an Extractive Electrospray Ionization Time-of-Flight Mass Spectrometer (EESI-TOF-MS) – Part 1: Biogenic Influences and Day–Night Chemistry in Summer. *Atmospheric Chemistry and Physics* **2019**, *19* (23), 14825–14848. https://doi.org/10.5194/acp-19-14825-2019.